# FreeSplatter: Pose-free Gaussian Splatting for Sparse-view 3D Reconstruction

## Abstract

Existing sparse-view reconstruction models heavily rely on accurate known camera poses. However, deriving camera extrinsics and intrinsics from sparse-view images presents significant challenges. In this work, we present **FreeSplatter**, a highly scalable, feed-forward reconstruction framework capable of generating high-quality 3D Gaussians from **uncalibrated** sparse-view images and recovering their camera parameters in mere seconds. FreeSplatter is built upon a streamlined transformer architecture, comprising sequential self-attention blocks that facilitate information exchange among multi-view image tokens and decode them into pixel-wise 3D Gaussian primitives. The predicted Gaussian primitives are situated in a unified reference frame, allowing for high-fidelity 3D modeling and instant camera parameter estimation using off-the-shelf solvers. To cater to both **object-centric** and **scene-level** reconstruction, we train two model variants of FreeSplatter on extensive datasets. In both scenarios, FreeSplatter outperforms state-of-the-art baselines in terms of reconstruction quality and pose estimation accuracy. Furthermore, we showcase FreeSplatter's potential in enhancing the productivity of downstream applications, such as text/image-to-3D content creation.

## 1 Introduction

Recent breakthroughs in neural scene representation and differentiable rendering, *e.g.*, Neural Radiance Fields (NeRF) (Mildenhall et al., 2021) and Gaussian Splatting (GS) (Kerbl et al., 2023), have shown unprecedented multi-view reconstruction quality for densely-captured images with calibrated camera poses, employing a per-scene optimization approach. However, they are not applicable to sparse-view scenarios, where classical camera calibration techniques like Structure-from-Motion (SfM) (Schonberger & Frahm, 2016) tend to fail due to insufficient image overlaps. Generalizable reconstruction models (Hong et al., 2024b; Xu et al., 2024a; Charatan et al., 2024) attempt to address sparse-view reconstruction in a feed-forward manner by utilizing learned data priors. Despite their efficiency and generalization capabilities, these models predominantly assume access to accurate camera poses and intrinsics, or imply that they are obtained in a pre-processing step, thereby circumventing the difficulty of deriving camera parameters in real application scenarios. Liberating sparse-view reconstruction from known camera poses remains a significant challenge.

Prior works have explored training *pose-free* reconstruction models. PF-LRM (Wang et al., 2024a) and LEAP (Jiang et al., 2024b) share a similar framework that maps multi-view image tokens to a NeRF representation with a transformer. Despite the milestone they have set, their NeRF representation suffers from inefficient volume rendering and low resolution, limiting the training efficiency and scalability to complex scenes. Besides, inferring the input camera poses from such implicit representations is not trivial. PF-LRM employs an additional branch to predict per-view coarse point clouds for camera pose prediction, which introduces extra training cost and difficulties. DUSt3R (Wang et al., 2024b) presents a novel paradigm for joint 3D reconstruction and pose estimation by modeling SfM and Multi-view Stereo (MVS) as an end-to-end point-regression task. By regressing the "point maps", *i.e.*, the 3D un-projection of depth maps, from a stereo image pair in a unified reference frame, it can recover the relative camera pose efficiently with a PnP (Fischler & Bolles, 1981; Hartley & Zisserman, 2003) solver. DUSt3R demonstrates impressive zero-shot 3D reconstruction and pose prediction capability by training on a broad range of datasets.

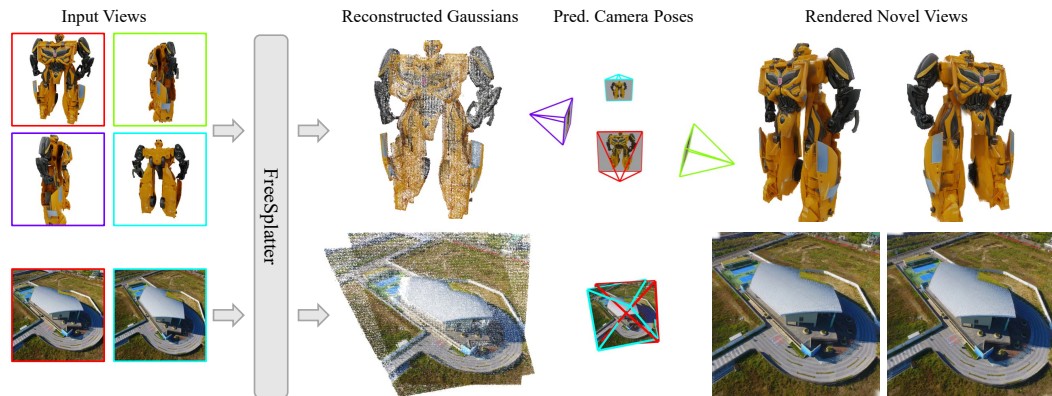

Figure 1: Given **uncalibrated** sparse-view images, our FreeSplatter can reconstruct pixel-wise 3D Gaussians, enabling both high-fidelity novel view rendering and instant camera pose estimation in mere seconds. FreeSplatter can deal with both object-centric (up) and scene-level (down) scenarios.

However, the sparsity of point cloud makes it a suboptimal representation for many downstream applications, *e.g.*, novel view synthesis. In comparison, 3D Gaussian Splats (3DGS) can encode high-fidelity radiance fields and enable efficient view synthesis by augmenting point clouds with additional Gaussian attributes. We then ask, can we directly predict the "Gaussian maps" from multi-view images to achieve both high-quality 3D modeling and instant camera pose estimation? In this work, we presents **FreeSplatter**, a feed-forward reconstruction framework that predicts per-pixel Gaussians from **uncalibrated** sparse-view images and estimates their camera parameters jointly, to say "yes" to this question. The core of FreeSplatter is a scalable single-stream transformer architecture that maps multi-view image tokens into pixel-aligned Gaussian maps with simple self-attention layers, requiring no input camera poses, intrinsics, nor post-alignment steps. The predicted Gaussian maps serve as a high-fidelity and efficient 3D scene representation, while the Gaussian locations enable ultra-fast camera extrinsics and intrinsics estimation in mere seconds using off-the-shelf solvers.

The training and rendering efficiency of 3D Gaussians makes it possible to consider more complex scene-level reconstruction. We thus train two FreeSplatter variants on Objaverse (Deitke et al., 2023) and a mixture of multiple scene datasets (Yao et al., 2020; Yeshwanth et al., 2023; Reizenstein et al., 2021) to handle object-centric and scene-level reconstruction scenarios, respectively. The two variants, denoted as **FreeSplatter-O** (object) and **FreeSplatter-S** (scene), share a common model architecture with ∼306 million parameters while making essential adjustments for task adaption. We conduct extensive experiments to demonstrate the superiority of FreeSplatter compared to prior sparse-view reconstruction methods in terms of reconstruction quality and pose estimation accuracy. To be highlighted, FreeSplatter-O outperforms previous *pose-dependent* large reconstruction models by a large margin, while FreeSplatter-S achieves a pose-estimation accuracy on par with MASt3R (Leroy et al., 2024), the upgraded version of DUSt3R, on ScanNet++ and CO3D benchmarks. Furthermore, we showcase the potential of FreeSplatter in enhancing the productivity of downstream applications, such as text/image-to-3D content creation.

## 2 RELATED WORK

**Large Reconstruction Models.** The availability of large-scale 3D datasets (Deitke et al., 2023; 2024) unlocks the possibility of training highly generalizable models for open-category image-to-3D reconstruction. Large Reconstruction Models (LRMs) (Hong et al., 2024b; Xu et al., 2024c; Li et al., 2024a) leverage a scalable feed-forward transformer architecture to map sparse-view image tokens into a 3D triplane NeRF (Mildenhall et al., 2021; Chan et al., 2022) and supervise it with multi-view rendering loss. Subsequent works have made efforts on adopting mesh (Xu et al., 2024a; Wei et al., 2024; Wang et al., 2024c) or 3D Gaussians representation (Tang et al., 2024; Xu et al., 2024b; Zhang et al., 2024b) for real-time rendering speed, exploring more efficient network architectures (Zheng et al., 2024; Zhang et al., 2024a; Chen et al., 2024a; Li et al., 2024b; Cao et al., 2024), improving the texture quality (Boss et al., 2024; Siddiqui et al., 2024; Yang et al., 2024a), and exploiting explicit 3D supervision for better geometry (Liu et al., 2024a). Despite the superior reconstruction quality

and generalization capability, LRMs require *posed* images as input and are highly sensitive to the pose accuracy, limiting their applicable scenarios.

**Pose-free Reconstruction.** Classical pose-free reconstruction algorithms like Structure from Motion (SfM) (Hartley & Zisserman, 2003; Ullman, 1979; Schonberger & Frahm, 2016) begins with finding pixel correspondences across multiple views first, and then perform 3D points triangulation and bundle adjustment to optimize 3D coordinates and camera parameters jointly. Recent efforts on enhancing the SfM pipeline encompass leveraging learning-based feature descriptors (DeTone et al., 2018; Dusmanu et al., 2019; Revaud et al., 2019; Wang et al., 2023c) and image matchers (Edstedt et al., 2023; 2024; Sarlin et al., 2020; Lindenberger et al., 2023), as well as differentiable bundle adjustment (Lin et al., 2021; Wang et al., 2023a; Wei et al., 2020). SfM-based methods works well with sufficient image overlaps between nearby views, but face challenges in matching correspondences from sparse views. On the other hand, learning-based methods (Jiang et al., 2024a;b; Hong et al., 2024a; Fan et al., 2023) rely on learned data priors to recover the 3D geometry from input views. PF-LRM (Wang et al., 2024a) extends the LRM framework by predicting a per-view coarse point cloud for camera pose estimation with a differentiable PnP solver (Chen et al., 2022). DUSt3R (Wang et al., 2024b) presents a novel framework for Multi-view stereo (MVS) by modeling it as a pointmap-regression problem. Subsequent works further improve it in local representation capability (Leroy et al., 2024) and reconstruction efficiency (Wang & Agapito, 2024).

**Generalizable Gaussian Splatting.** Compared to NeRF's (Mildenhall et al., 2021) MLP-based implicit representation, 3DGS represents a scene explicitly with a point cloud, which strikes the balance between high-fidelity and real-time rendering speed. However, the per-scene optimization of 3DGS requires densely-captured images and sparse point cloud generated by SfM for initialization. Recent works (Charatan et al., 2024; Chen et al., 2024b; Szymanowicz et al., 2024; Liu et al., 2024b; Wewer et al., 2024) have explored feed-forward models for sparse-view Gaussian reconstruction by capitalizing on large-scale datasets and scalable model architectures. These models typically assume the access to accurate camera poses and utilize 3D-to-2D geometric projection for feature aggregation, adopting model designs like epipolar lines (Charatan et al., 2024) or plane-swept cost-volume (Chen et al., 2024b; Liu et al., 2024b). InstantSplat (Fan et al., 2024) and Splatt3R (Smart et al., 2024) leverage the pose-free reconstruction capability of DUSt3R/MASt3R for sparse-view reconstruction. The former initializes the Gaussian positions using the DUSt3R point cloud and then optimizes other Gaussian paremters, while the latter trains a Gaussian head in a frozen MASt3R model. Despite the impressive results, their reconstruction quality highly depends on the point cloud quality generated by DUSt3R.

## 3 METHOD

Given a sparse set of $N$ input images $\{\boldsymbol{I}^n \mid n = 1, \ldots, N\}$ with no known camera extrinsics nor intrinsics, FreeSplatter aims to reconstruct the scene as a set of Gaussians and estimate the camera parameters of the $N$ images simultaneously. The pipeline can be formulated as:

$$\boldsymbol{G}, \boldsymbol{P}^1, \ldots, \boldsymbol{P}^N, f = \text{FreeSplatter}\left(\boldsymbol{I}^1, \ldots, \boldsymbol{I}^N\right), \tag{1}$$

where $\boldsymbol{G}$ denotes the reconstructed Gaussians which is the union of $N$ pixel-wise Gaussian maps, *i.e.*, $\boldsymbol{G} = \{\boldsymbol{G}^n \mid n = 1, \ldots, N\}$, $\boldsymbol{P}^n$ denotes the estimated camera pose for $\boldsymbol{I}^n$, and $f$ denotes the focal length. We assume a common focal length for all input images which is reasonable in most scenarios.

We implement FreeSplatter as a feed-forward transformer that takes multi-view image as input and predict $N$ Gaussian maps $\{\boldsymbol{G}^n \mid n = 1, \ldots, N\}$ in a unified reference frame. The Gaussian maps enable both high-fidelity scene modeling and ultra-fast camera parameter estimation using off-the-shelf solvers (Fischler & Bolles, 1981; Hartley & Zisserman, 2003; Plastria, 2011), since the Gaussian centers have located the surface point cloud explicitly. In this section, we first briefly review the background knowledge of Gaussian Splatting (Section 3.1), and then introduce the model architecture (Section 3.2) and training details (Section 3.3) of our approach.

### 3.1 PRELIMINARY

3D Gaussian Splatting (3DGS) (Kerbl et al., 2023) represents a scene as a set of 3D Gaussian primitives $\{\boldsymbol{g}_k = (\boldsymbol{\mu}_k, \boldsymbol{r}_k, \boldsymbol{s}_k, o_k, \boldsymbol{c}_k) \mid k = 1, \ldots, K\}$, each primitive is parameterized by location

Figure 2: **FreeSplatter pipeline.** Given input views $\{\boldsymbol{I}^n \mid n = 1, \ldots, N\}$ without any known camera extrinsics or intrinsics, we first patchify them into image tokens, and then feed all tokens into a sequence of self-attention blocks to exchange information among multiple views. Finally, we decode the output image tokens into $N$ Gaussian maps $\{\boldsymbol{G}^n \mid n = 1, \ldots, N\}$, from which we can render novel views, as well as recovering camera focal length $f$ and poses $\{\boldsymbol{P}^n \mid n = 1, \ldots, N\}$ with simple iterative solvers.

$\boldsymbol{\mu}_k \in \mathbb{R}^3$, rotation quaternion $\boldsymbol{r}_k \in \mathbb{R}^4$, scale $\boldsymbol{s}_k \in \mathbb{R}^3$, opacity $o_k \in \mathbb{R}$, and Spherical Harmonic (SH) coefficients $\boldsymbol{c}_k \in \mathbb{R}^{3 \times d}$ for computing view–dependent color, with $d$ denoting the degree of SH. These primitives parameterize the scene's radiance field with an explicit point cloud and can be rendered into novel views with an efficient rasterizer. Compared to the expensive volume rendering process of NeRF, 3DGS achieves comparable rendering quality while being much more efficient in terms of time and memory.

## 3.2 Model Architecture

As Figure 2 shows, FreeSplatter employs a transformer architecture similar to GS-LRM (Zhang et al., 2024b). Given a set of uncalibrated images $\{\boldsymbol{I}^n \mid n = 1, \ldots, N\}$, we first tokenize them into image tokens $\{\boldsymbol{e}^{n,m} \mid n = 1, \ldots, N, m = 1, \ldots, M\}$, where $M$ denotes the patch number of each image. Then we feed all image tokens into a sequence of self-attention blocks to exchange information among multiple views, and finally decode them into $N$ Gaussian maps $\{\boldsymbol{G}^n \mid n = 1, \ldots, N\}$ composed of per-pixel Gaussian primitives, from which we can render novel views and recover camera parameters with simple iterative solvers.

**Image Tokenization.** The only input of our model is $N$ images $\{\boldsymbol{I}^n \in \mathbb{R}^{H \times W \times 3} \mid n = 1, \ldots, N\}$. Following ViT (Dosovitskiy et al., 2021), we split the images into $p \times p$ patches ($p = 8$) and flatten them into 1D vectors of dimension $p^2 \cdot 3$. Then a linear layer is applied to map the vectors into $d$-dim image tokens ($d$ equals to the transformer width). Denoting the $m$-th token of the $n$-th image as $\boldsymbol{e}^{n,m}$, we add it with two additional embeddings before feeding it into the transformer:

$$\boldsymbol{e}^{n,m} = \boldsymbol{e}^{n,m} + \boldsymbol{e}^m_{\text{pos}} + \boldsymbol{e}^n_{\text{view}}, \tag{2}$$

where $\boldsymbol{e}^m_{\text{pos}}$ denotes the position embedding for the $m$-th patch, and $\boldsymbol{e}^n_{\text{view}}$ is a view embedding for distinguishing the reference view and other views. Specifically, **we take the first image as the reference view and predict all Gaussian in its camera frame**. We use a reference embedding $\boldsymbol{e}^{\text{ref}}$ for the first view ($n = 1$) and another source embedding $\boldsymbol{e}^{\text{src}}$ for other views ($n = 2, \ldots, N$), both of which are learnable during training.

**Feed-forward Transformer.** The image tokens added with position and view embeddings are then fed into a sequence of $L$ self-attention blocks to exchange information across multiple views. Each block consists of a self-attention layer and an MLP layer, equipping with pre-normalization and residual connection (He et al., 2016). The forward process of each block can be formulated as:

$$\boldsymbol{e}^{n,m}_{\text{attn}} = \text{SelfAttn}\left(\text{LayerNorm}\left(\boldsymbol{e}^{n,m}_{\text{in}}\right), \{\text{LayerNorm}\left(\boldsymbol{e}^{n,m}_{\text{in}}\right)\}\right) + \boldsymbol{e}^{n,m}_{\text{in}},$$
$$\boldsymbol{e}^{n,m}_{\text{out}} = \text{MLP}\left(\text{LayerNorm}\left(\boldsymbol{e}^{n,m}_{\text{attn}}\right)\right) + \boldsymbol{e}^{n,m}_{\text{attn}}. \tag{3}$$

**Gaussian Map Prediction.** For each output token $\boldsymbol{e}^{n,m}_{\text{out}}$ of the last ($L$-th) block, we decode it back into $p^2$ Gaussians with a simple linear layer, leading to a 1D vector of dimension $p^2 \cdot q$, where $q$

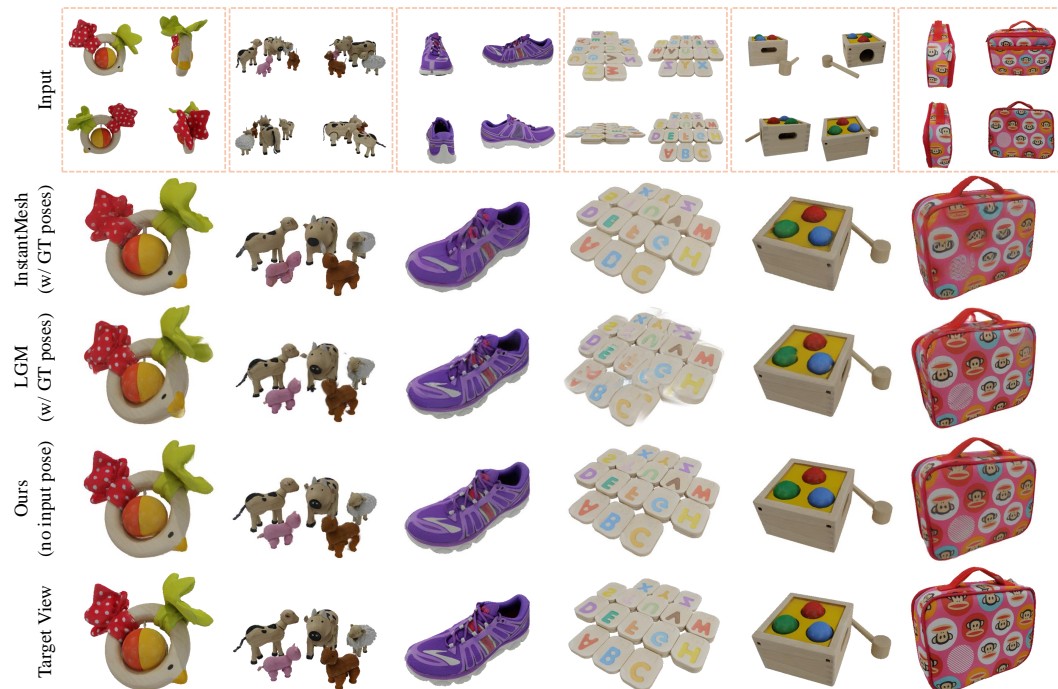

Figure 3: **Object-centric Sparse-view Reconstruction.** We show 6 samples from the Google Scanned Objects dataset. To be noted, the results of LGM and InstantMesh (2nd and 3rd rows) are generated with ground truth camera poses (and intrinsics), while our results (4th row) are generated in a completely pose-free manner.

is the parameter number of each Gaussian primitive. Then we unpatchify the predicted Gaussian parameters into a Gaussian patch $G^{n,m} \in \mathbb{R}^{p \times p \times q}$. Finally, we spatially concatenate the Gaussian patches $G^{n,m}$ for each view, leading to $N$ Gaussian maps $\left\{ G^n \in \mathbb{R}^{H \times W \times q} \mid n = 1, \ldots, N \right\}$.

Each pixel in the Gaussian maps contains a $q$-dim 3D Gaussian primitive. Prior pose-dependent Gaussian LRMs (Zhang et al., 2024b; Xu et al., 2024b; Tang et al., 2024) parameterize each Gaussian's location with a single depth value. However, our model assumes no available camera poses nor instrinsics, making it impossible to un-project the Gaussian locations from depths. Therefore, we directly predict the actual Gaussian locations in the reference frame and adopt a pixel-alignment loss to restrict Gaussians to lie on camera rays (see Section 3.3 for details).

**Camera Pose Estimation.** To recover the camera poses, we first estimate the focal length $f$ from predicted Gaussian maps (see Section A.1 for details). Different from DUSt3R that can only feed a pair of images into the model to predict pair-wise point maps and requires an additional alignment to unify the pair-wise reconstruction results into a global point cloud, FreeSplatter predicts the Gaussian maps of all views in a unified reference frame, thus can recover the camera poses $\{P^n \mid n = 1, \ldots, N\}$ in a feed-forward manner without global alignment. For each Gaussian location map $X^n \in \mathbb{R}^{H \times W \times 3}$ (the first 3 channels of $G^n$), its pixel coordinate map $Y^n \in \mathbb{R}^{H \times W \times 2}$ and a valid mask $M^n \in \mathbb{R}^{H \times W}$, we apply a PnP-RANSAC solver (Hartley & Zisserman, 2003; Bradski, 2000) to estimate $P^n \in \mathbb{R}^{4 \times 4}$:

$$P^n = \text{PnP}\left(X^n, Y^n, M^n, K\right), \tag{4}$$

where $K = [[f, 0, \frac{W}{2}], [0, f, \frac{H}{2}], [0, 0, 1]]$ denotes the estimated intrinsic matrix. The mask $M^n$ aims to mark the pixel locations that contribute to the pose optimization, which differs in object-centric and scene-level reconstruction scenarios, please see Section A.1 for more details.

### 3.3 TRAINING DETAILS

We train two FreeSplatter variants targeting for object-centric and scene-level pose-free reconstruction. They share the same model architecture and parameter size, but we made essential adjustments in training objectives and strategies to adapt them to different scenarios.

**Staged Training Strategy.** Prior pose-dependent LRMs (Hong et al., 2024b; Li et al., 2024a; Xu et al., 2024b; Zhang et al., 2024b) leverage pure rendering loss for supervision. However, our model assumes no known camera poses nor intrinsics and the Gaussian positions are free in 3D space, making it extremely challenging to predict correct Gaussian positions. Gaussian-based reconstruction approaches heavily rely on the initialization of Gaussian positions, *e.g.*, 3DGS (Kerbl et al., 2023) initializes the Gaussian positions with the sparse point cloud generated by SfM, while the parameters of our model are randomly initialized at the beginning. In practice, we found it essential to supervise the Gaussian positions at the beginning:

$$\mathcal{L}_{\text{pos}} = \sum_{n=1}^{N} \left\| \boldsymbol{M}^n \odot \hat{\boldsymbol{X}}^n - \boldsymbol{M}^n \odot \boldsymbol{X}^n \right\|, \tag{5}$$

where $\hat{\boldsymbol{X}}^n \in \mathbb{R}^{H \times W \times 3}$ is the predicted Gaussian position map, $\boldsymbol{X}^n$ is the ground truth point map unprojected from depths. $\boldsymbol{M}^n \in \mathbb{R}^{H \times W}$ is a mask indicating the pixels storing valid depth values, which is exactly the foreground object mask for object-centric reconstruction. For scene-level reconstruction, $\boldsymbol{M}^n$ depends on where the depth values are defined in different datasets.

We apply $\mathcal{L}_{\text{pos}}$ in the pre-training stage, so that the model learns to predict approximately correct Gaussian positions. In our experiments, this pre-training is **essential** to model's convergence. However, $\mathcal{L}_{\text{pos}}$ can only supervise the pixels with valid depths, while the Gaussian positions predicted at other pixels remain unconstrained. Besides, the ground truth depths are noisy in some datasets, and applying $\mathcal{L}_{\text{pos}}$ throughout the training leads to degraded rendering quality. To provide a more stable geometric supervision, we adopt a pixel-alignment loss to enforce each predicted Gaussian to be aligned with its corresponding pixel, which maximizes the ray cosine similarities:

$$\mathcal{L}_{\text{align}} = \sum_{n=1}^{N} \sum_{i=0}^{H} \sum_{j=0}^{W} \left( 1 - \frac{\hat{\boldsymbol{r}}_{i,j}^n \cdot \boldsymbol{r}_{i,j}^n}{\|\hat{\boldsymbol{r}}_{i,j}^n\| \|\boldsymbol{r}_{i,j}^n\|} \right), \tag{6}$$

where $\boldsymbol{r}_{i,j}^n$ denotes the ray from the camera origin $\boldsymbol{t}^n$ to point $\boldsymbol{X}_{i,j}^n$. $\mathcal{L}_{\text{align}}$ restricts the predicted Gaussians to be distributed on the camera rays, which improves the rendering quality. Besides, enforcing the pixel-alignment is important to camera parameter estimation, since the iterative solvers minimize the pixel-projection errors.

**Loss Functions.** The overall training objective of our model is:

$$\mathcal{L} = \mathcal{L}_{\text{render}} + \lambda_{\text{a}} \cdot \mathcal{L}_{\text{align}} + \mathbf{1}_{\text{t} \leq \text{T}_{\max}} \lambda_{\text{p}} \cdot \mathcal{L}_{\text{pos}}, \tag{7}$$

where the rendering loss $\mathcal{L}_{\text{render}}$ is a combination of MSE and LPIPS loss. $t$ and $T_{\max}$ denote the current training step and maximum pre-training step, respectively. In our implementation, we set $\lambda_{\text{a}} = 1.0, \lambda_{\text{p}} = 10.0$.

**Occlusion in Pixel-aligned Gaussians.** Prior pose-dependent Gaussian-basd LRMs (Tang et al., 2024; Zhang et al., 2024b; Xu et al., 2024b) parameterize the Gaussian positions with a single depth value to ensure they are pixel-aligned. Despite the simplicity, pixel-aligned Gaussians can only represent the areas captured by input images. It is challenging for sparse input views to cover the whole scene, resulting in missing reconstruction results in the occluded areas. Our model also suffers from this problem due to the usage of $\mathcal{L}_{\text{align}}$. For the object-centric model, we adopt a simple strategy to alleviate this problem by only calculate Equation 6 in the foreground area, while allowing other Gaussians to move freely and model the occluded areas.

For the scene-level model trained on real-world images, all predicted Gaussians have be pixel-aligned to model the complex background. We instead focus on reconstructing the observed areas. To avoid the areas in the target views that are invisible in input views provide negative guidance for training, we adopt the target-view masking strategy proposed by Splatt3R and only calculate the rendering loss in visible areas. Please refer to Smart et al. (2024) for more details.

## 4 EXPERIMENTS

Due to the space limit, we put more comprehensive experimental results in Section A.2 of the appendix, including comparison on object-level reconstruction with PF-LRM (Wang et al., 2024a) (Section A.2.1), more extensive image-to-3D generation results with FreeSplatter (Section A.2.2), cross-dataset generalization results (Section A.2.3), and additional ablation studies (Section A.3).

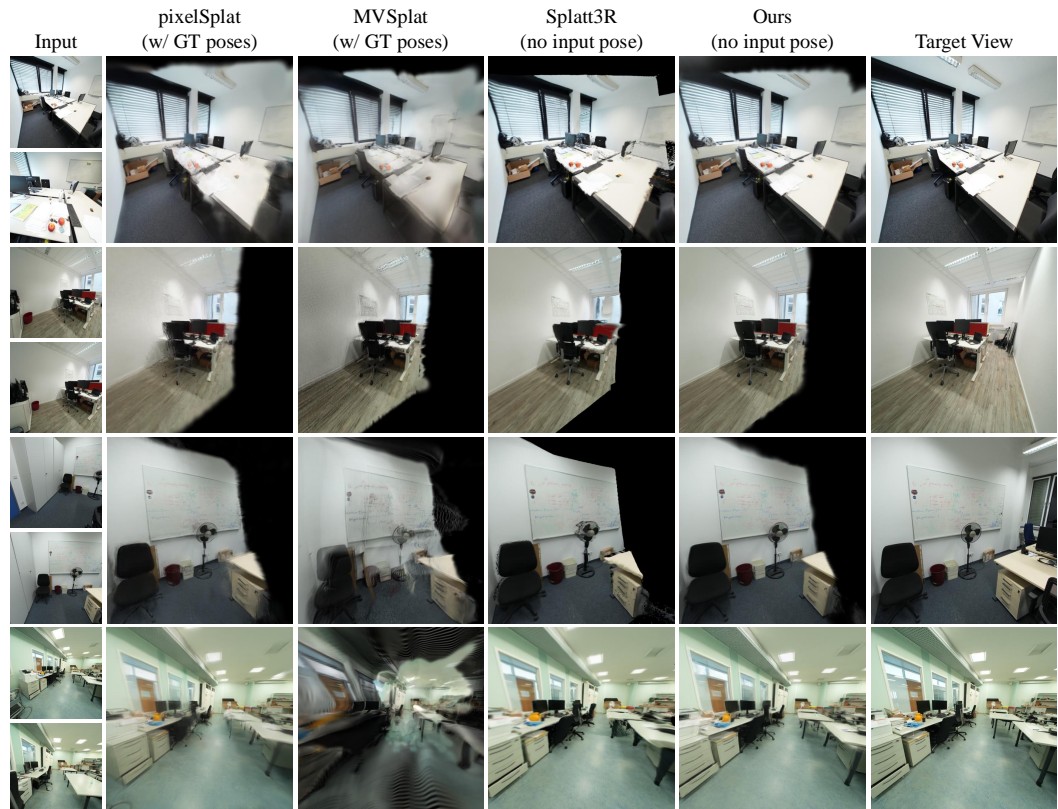

Figure 4: **Scene-level Reconstruction on ScanNet++.** The results of pixelSplat and MVSplat are obtained with ground truth input poses, while the results of Splat3R and ours are pose-free.

### 4.1 EXPERIMENTAL SETTINGS

**Training Datasets.** FreeSplatter-O assumes the input images are white-background with an object in the center, which is trained on Objaverse (Deitke et al., 2023), a large-scale dataset containing around 800K 3D assets. For each 3D asset, we normalize it into a $[-1, 1]^3$ cube, and then render 32 random views associated with depth maps around it, all in a resolution of $512 \times 512$. FreeSplatter-S is trained on a mixture of multiple datasets, including BlendedMVS (Yao et al., 2020), ScanNet++ (Yeshwanth et al., 2023) and CO3Dv2 (Reizenstein et al., 2021), which is a subset of DUSt3R's (Wang et al., 2024b) training datasets covering the most representative scene types: outdoor, indoor, and realistic objects.

**Evaluation Datasets.** For object-centric reconstruction and pose-estimation, we evaluate our model on OmniObject3D (Wu et al., 2023) and Google Scanned Objects (GSO) (Downs et al., 2022) datasets, containing around 6K and 1K scanned 3D assets, respectively. We use the whole GSO dataset and pick out 300 OmniObject3D objects in 30 categories as the evaluation set. For each object, we render 20 random views and 4 additional structured input views for the sparse-view reconstruction task, which are evenly placed around the objects with an elevation of $20°$ to ensure a good coverage. For scene-level reconstruction and pose estimation, we mainly evaluate on the test splits of ScanNet++ (Yeshwanth et al., 2023) and CO3Dv2 (Reizenstein et al., 2021).

### 4.2 SPARSE-VIEW RECONSTRUCTION

**Baselines.** Prior pose-free object reconstruction models like LEAP (Jiang et al., 2024b) were only trained on small datasets, showing very limited generalization capability in open-domain datasets. PF-LRM (Wang et al., 2024a) is the most comparable work to our FreeSplatter-O. Although its code is not publicly available, we compare with its inference results on its evaluation datasets on both sparse-view reconstruction and camera pose estimation tasks. Due to the space limit, we put the results in Section A.2.1 of the appendix, which demonstrate that FreeSplatter-O outperforms PF-

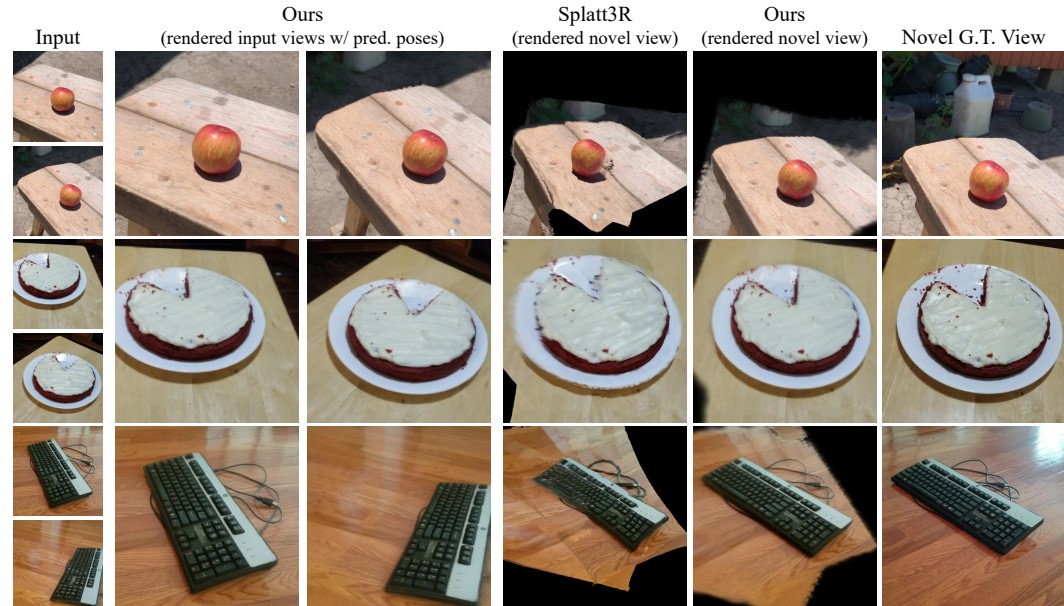

Figure 5: **Scene-level Reconstruction on CO3Dv2.**

**LRM on its OmniObject3D benckmark and preserves visual details better.** We also compare with two pose-dependent large reconstruction models, *i.e.*, LGM (Tang et al., 2024) and InstantMesh (Xu et al., 2024a), which are base on 3D Gaussians and tri-plane NeRF, respectively. We directly feed ground truth camera poses to them, while our model is totally pose-free.

For scene-level reconstruction, we compare with two state-of-the-art works on generalizable Gaussians, *i.e.*, pixelSplat (Charatan et al., 2024) and MVSplat (Chen et al., 2024b), both of which require camera poses. We fine-tune their checkpoints trained on RealEstate10K (Zhou et al., 2018) on Scan-Net++ for more fair comparison, while do not apply them on CO3Dv2 due to significant domain gap. Besides, we also compare with an up-to-date pose-free Gaussian reconstruction method, *i.e.*, Splatt3R (Smart et al., 2024), which utilizes frozen MASt3R (Leroy et al., 2024) model for Gaussian location prediction, while fine-tuning an additional head for predicting other Gaussian attributes.

**Metrics.** For both object-centric and scene-level reconstruction, We evaluate the rendering quality of reconstructed Gaussians and report the standard metrics for novel view synthesis, *i.e.*, PSNR, SSIM and LPIPS. All metrics are evaluated at the resolution of $512 \times 512$.

**Results and Analysis.** As demonstrated in Table 2, FreeSplatter models achieve superior performance across both object-centric and scene-level reconstruction tasks without camera poses. FreeSplatter-O significantly outperforms LGM and InstantMesh across all metrics even if they are pose-dependent, achieving PSNR improvements of $> 5$ and $> 7$ on previously unseen GSO and OmniObject3D datasets, respectively. Qualitative comparisons in Figure 3 further illustrate this performance gap, with InstantMesh and LGM exhibiting blurry artifacts (notably in the alphabets, 4th column), while our approach maintains clarity and detail fidelity consistent with ground truth views.

Existing Large Reconstruction Models (LRMs) assume the necessity of accurate camera poses for high-quality 3D reconstruction, incorporating pose information through LayerNorm modulation (Li et al., 2024a) or plucker ray embeddings (Xu et al., 2024c; Tang et al., 2024; Xu et al., 2024b). However, FreeSplatter-O's superior performance challenges this assumption, suggesting that **camera poses may not be essential for developing high-quality, scalable reconstruction models**.

In scene-level reconstruction on ScanNet++, FreeSplatter-S demonstrates superior performance across most metrics compared to pose-dependent pixelSplat and MVSplat, highlighting its robust reconstruction quality and generalization capability. While the pose-free competitor Splatt3R leverages MASt3R's (Leroy et al., 2024) 3D point regression capabilities trained on extensive datasets, its performance is constrained by MASt3R's prediction errors due to its frozen architecture and Gaussian head-only training, resulting in fixed Gaussian positions. In contrast, our end-to-end training approach with rendering supervision enables simultaneous optimization of Gaussian locations and attributes. Qualitative results on ScanNet++ and CO3Dv2 (Figure 4, Figure 5) illustrate

Table 1: **Quantitative results on camera pose estimation.** We highlight the best metric as red.

| Method | OmniObject3D | | | | GSO | | | |
|---|---|---|---|---|---|---|---|---|
| | RRE ↓ | RRA@15° ↑ | RRA@30° ↑ | TE↓ | RRE ↓ | RRA@15° ↑ | RRA@30° ↑ | TE↓ |
| FORGE | 76.822 | 0.081 | 0.257 | 0.430 | 97.814 | 0.022 | 0.083 | 0.898 |
| MASt3R | 96.670 | 0.052 | 0.112 | 0.524 | 61.820 | 0.244 | 0.445 | 0.353 |
| FreeSplatter-S | 83.795 | 0.076 | 0.167 | 0.631 | 72.914 | 0.212 | 0.396 | 0.426 |
| FreeSplatter-O | 11.550 | 0.909 | 0.937 | 0.081 | 3.851 | 0.962 | 0.978 | 0.030 |
| Method | ScanNet++ | | | | CO3Dv2 | | | |
| | RRE ↓ | RRA@15° ↑ | RRA@30° ↑ | TE↓ | RRE ↓ | RRA@15° ↑ | RRA@30° ↑ | TE↓ |
| PoseDiffusion | - | - | - | - | 7.950 | 0.803 | 0.868 | 0.328 |
| MASt3R | 0.724 | 0.988 | 0.993 | 0.104 | 2.918 | 0.975 | 0.989 | 0.112 |
| FreeSplatter-S | 0.791 | 0.982 | 0.987 | 0.110 | 3.054 | 0.976 | 0.986 | 0.148 |

Table 2: **Quantitative results on sparse-view reconstruction.**

| Method | OmniObject3D | | | GSO | | |
|---|---|---|---|---|---|---|
| | PSNR ↑ | SSIM ↑ | LPIPS ↓ | PSNR ↑ | SSIM ↑ | LPIPS ↓ |
| LGM (w/ gt pose) | 24.852 | 0.942 | 0.060 | 24.463 | 0.891 | 0.093 |
| InstantMesh (w/ gt pose) | 24.077 | 0.945 | 0.062 | 25.421 | 0.891 | 0.095 |
| Ours (FreeSplatter-O) | 31.929 | 0.973 | 0.027 | 30.443 | 0.945 | 0.055 |
| Method | ScanNet++ | | | CO3Dv2 | | |
| | PSNR ↑ | SSIM ↑ | LPIPS ↓ | PSNR ↑ | SSIM ↑ | LPIPS ↓ |
| pixelSplat (w/ gt pose) | 24.974 | 0.889 | 0.180 | - | - | - |
| MVSplat (w/ gt pose) | 22.601 | 0.862 | 0.208 | - | - | - |
| Splatt3R | 21.013 | 0.830 | 0.209 | 18.074 | 0.740 | 0.197 |
| Ours (FreeSplatter-S) | 25.807 | 0.887 | 0.140 | 20.405 | 0.781 | 0.162 |

our method's superior visual fidelity. To be noted, novel view synthesis for both FreeSplatter and Splatt3R is performed by aligning predicted cameras with target viewpoints.

### 4.3 CAMERA POSE ESTIMATION

**Baselines.** To evaluate our method's pose estimation capabilities, we benchmark against MASt3R, the current state-of-the-art in zero-shot multi-view pose estimation. Additional comparisons include FORGE (Jiang et al., 2024a) for object-centric datasets and PoseDiffusion (Fan et al., 2023) for scene-level reconstruction, with PoseDiffusion evaluated exclusively on CO3Dv2 due to its training scope. Traditional COLMAP-based approaches (Schonberger & Frahm, 2016) are excluded from our evaluation due to their documented high failure rates in sparse-view scenarios, a limitation consistently reported in previous studies (Wang et al., 2024a).

**Metrics.** Following established protocols Wang et al. (2024a; 2023b), we evaluate pose estimation through distinct rotation and translation metrics. Rotational accuracy is assessed via relative rotation error (RRE) in degrees and relative rotation accuracy (RRA), where RRA represents the percentage of rotation errors below predefined thresholds (15° and 30°). Translational accuracy is measured through translation error (TE) between predicted and ground truth camera centers. For scenarios with more than two input views, errors are computed pairwise and averaged across all camera combinations.

As shown in Table 1, FreeSplatter-O demonstrates substantial performance improvements over existing baselines on object-centric datasets, while FreeSplatter-S achieves comparable results with MASt3R on scene-level datasets. MASt3R's poor performance on rendered object-centric datasets without backgrounds can be attributed to the significant domain gap between its training data and these synthetic images. Specifically, rendered object-centric images lack the complex backgrounds and rich textures present in natural scenes, presenting unique challenges for pose estimation without adequate data priors. In scene-level evaluations on ScanNet++ and CO3Dv2, while MASt3R achieves marginally superior metrics, FreeSplatter-S maintains competitive performance despite its more limited training scope. This performance differential primarily stems from dataset scale disparity—our FreeSplatter-S utilizes only three datasets, a subset of DUSt3R/MASt3R's extensive training corpus, resulting in fewer learned data priors. Future work will focus on scaling the model across broader datasets to bridge this gap.

### 4.4 ABLATION STUDIES

**Number of Input Views.** We make an experiment on a GSO sample to illustrate how the number of input views influences the reconstruction quality on the object-centric scenarios. Please refer to Figure 16 in the appendix for more details.

Table 3: **Ablation on the pixel-alignment loss.**

| Method | GSO | | | Method | ScanNet++ | | |
|---|---|---|---|---|---|---|---|
| | PSNR↑ | SSIM↑ | LPIPS↓ | | PSNR↑ | SSIM↑ | LPISP↓ |
| FreeSplatter-O (w/o $\mathcal{L}_{\text{align}}$) | 26.684 | 0.898 | 0.092 | FreeSplatter-S (w/o $\mathcal{L}_{\text{align}}$) | 21.330 | 0.832 | 0.201 |
| FreeSplatter-O | 30.443 | 0.945 | 0.055 | FreeSplatter-S | 25.807 | 0.887 | 0.140 |

Figure 6: **3D content creation with FreeSplatter.** 1st and 2nd rows: Image-to-3D results using Zero123++ (input image, Gaussian visualization, two novel views). 3rd row: Text-to-3D results using MVDream (prompt shown above; two Gaussian visualizations, two novel views).

**Impact of Pixel-Alignment Loss.** We study the impact of pixel-alignment loss defined in Equation 6. We remove the pixel-alignment loss from the training of both models, and report the novel view rendering metrics on GSO and ScanNet++ dataset. We can observe significant drops in all metrics without the pixel-alignment loss. We also compare the visual results in Figure 15 of the appendix, demonstrating that applying pixel-alignment loss leads to higher-fidelity renderings.

## 4.5 APPLICATIONS

FreeSplatter can be seamlessly integrated into 3D content creation pipelines, and its pose-free nature can potentially bring great convenience to users and enhance the productivity by a large margin. In a classical 3D generation pipeline (Li et al., 2024a; Xu et al., 2024a; Tang et al., 2024; Wang et al., 2024c) that first generates multi-view images with a multi-view diffusion model and then feed them into an LRM for reconstruction, we have to figure out how the camera poses of the multi-view diffusion model are defined and carefully align them with the LRM. With FreeSplatter as the content creator, these tedious steps are not required any longer. We just feed the multi-view images into FreeSplatter and get the outputs instantly after several seconds. Figure 6 shows some results of text/image-to 3D generation with MVDream (Shi et al., 2024) and Zero123++ (Shi et al., 2023), respectively. We notice that our method can render significantly more clear views from the images than previous pose-dependent LRMs, showing stronger robustness to multi-view inconsistency.

## 5 CONCLUSION

In this work, we present FreeSplatter, a highly scalable framework for pose-free sparse-view reconstruction. By leveraging a single-stream transformer architecture and predicting multi-view Gaussian maps in a unified frame, FreeSplatter enables both high-fidelity 3D modeling and instant camera pose estimation. We provide two model variants for both object-centric and scene-level reconstruction, both showing superior recontruction qulity and pose estimation accuracy. FreeSplatter also exhibits great potential in enhancing the productivity of downstream applications like text/image-to-3D content creation, which can liberate users from dealing with complex camera poses.

**Limitations.** Despite the promising performance of FreeSplatter, its pre-training stage relies on depth data, making it non-trivial to be trained on datasets with no depth labels, *e.g.*, RealEstate10K and MVImgNet (Yu et al., 2023). Besides, FreeSplatter requires two distinct model variants to deal with object-centric and scene-level reconstructions, while a unified model for both tasks would be preferred. We leave this as the future work.

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

## A    APPENDIX

### A.1    IMPLEMENTATION DETAILS

**Implementation Details.** FreeSplatter-O and FreeSplatter-S share the same model architecture of a 24-layer transformer with a width of 1024, containing $\sim 306$ million parameters. We do not use bias term throughout the model except for the output linear layer for Gaussian prediction. The model is trained on $256 \times 256$ input images for 500K steps and then fine-tuned on $512 \times 512$ input images for 100K steps on 16 NVIDIA H800 GPUs. We adopt an AdamW (Loshchilov, 2017) optimizer with $\beta_1 = 0.9, \beta_2 = 0.95$, the weight decay and learning rate are set to 0.01 and $4 \times 10^{-5}$, respectively. The maximum pre-training step $T_{\max} = 2 \times 10^5$, *i.e.*, we adopt the Gaussian position loss in Equation 5 for the first 200K training steps. To enhance the training efficiency, we utilize the xFormers (Lefaudeux et al., 2022) library for memory-efficient attention, gradient checkpointing (Chen et al., 2016), bfloat16 training (Micikevicius et al., 2017), and deferred back-propagation (Zhang et al., 2022) for GS rendering. In each batch, We sample 4 and 2 input views for FreeSplatter-O and FreeSplatter-S respectively and 4 novel views for supervision.

**Camera Intrinsic Estimation.** Following DUSt3R (Wang et al., 2024b), we assume centered principle point and square pixels in this work. Thanks to the point-cloud-based Gaussian representation, we can extract the Gaussian locations as "point maps" $\left\{ \boldsymbol{X}^n \in \mathbb{R}^{H \times W \times 3} \mid n = 1, \dots, N \right\}$ from the predicted Gaussian maps. Since all points are predicted in the first view's camera frame, we can estimate the focal length of the first view by minimizing the pixel-projection errors:

$$f_*^1 = \arg\min_{f^1} \sum_{i=0}^{W} \sum_{j=0}^{H} \left\| (i', j') - f^1 \frac{\left( X_{i,j,0}^1, X_{i,j,1}^1 \right)}{X_{i,j,2}^1} \right\|, \tag{8}$$

where $(i', j') = (i - \frac{W}{2}, j - \frac{H}{2})$ denotes pixel coordinates relative to the principle point $\left( \frac{H}{2}, \frac{W}{2} \right)$. This optimization problem can be solved by Weiszfeld algorithm (Plastria, 2011) efficiently.

Different from DUSt3R that only accepts binocular images as input, our model architecture naturally supports single-view reconstruction. Thus we feed each image $\boldsymbol{I}^n$ into the model individually to predict its monocular Gaussian map $\boldsymbol{G}_{\mathrm{mono}}^n$, and then estimate its focal length $f^n$ from $\boldsymbol{X}_{\mathrm{mono}}^n$. Finally, we compute the average focal length and adopt it for all views: $f = \frac{1}{N} \sum_{n=1}^{N} f^n$.

**Masks for PnP-RANSAC solver.** We adopt different masks $\boldsymbol{M}^n$ in Equation 4 for object-centric and scene-level reconstruction when estimating the input camera poses:

- For object-centric reconstruction, the model is trained on white-background images rendered from 3D assets, and the predicted Gaussians at the background area are not restricted to be pixel-aligned (See Section 3.3 for details), which may influence the camera pose solving. Therefore, we assume white-background input images at inference time, and adopt the foreground mask segmented by an off-the-shelf background-removal tool[1] as $\boldsymbol{M}^n$. The segmentation masks are not always perfect but are precise enough for the PnP-RANSAC solver to estimate the poses robustly.

- For scene-level reconstruction, we do not distinguish the foreground and background areas during training, and all the predicted Gaussians are restricted to be pixel-aligned by applying the pixel-alignment loss universally. Therefore, all visible Gaussians should contribute to pose solving and we use the Gaussian opacity map $\boldsymbol{O}^n \in \mathbb{R}^{H \times W}$ to compute $\boldsymbol{M}^n$ with a minimal visibility threshold $\tau$: $\boldsymbol{M}^n = (\boldsymbol{O}^n > \tau)$.

**Camera Normalization.** The definition of reconstruction frame plays an important role in model training. We take the camera frame of the first view as the reference frame for reconstruction, and normalize all cameras in a training batch into this frame. Besides, a scaling operation is required to deal with various scene scales. Denoting the original input cameras as $\left\{ \boldsymbol{P}_{\mathrm{in}}^n = [\boldsymbol{R}_{\mathrm{in}}^n \mid \boldsymbol{t}_{\mathrm{in}}^n] \mid n = 1, \dots, N \right\}$, we adopt different camera normalization strategies for object-centric and scene-level reconstruction.

For object-centric reconstruction, the training cameras are sampled orbiting the object center. We first scale all input cameras to make the distance from the first camera to the object center equal to

---

[1] https://github.com/danielgatis/rembg

2, and then transform all cameras into the reference frame so that the first view's camera pose is an identity matrix. With this strategy, we fix the object center at $(0, 0, 2)$ in the reference frame. For scene-level reconstruction, the camera distributions are more complex. Thus we fist transform all cameras into the reference frame, and then scale them using a factor $s = 1/d$, $d$ is the mean distance of all valid points to the origin:

$$d = \frac{1}{\sum_{n=1}^{N} |\boldsymbol{M}^n|} \sum_{n=1}^{N} \sum_{i=0} \sum_{j=0} M_{i,j}^n \cdot \left\| \boldsymbol{X}_{i,j}^n \right\|, \tag{9}$$

where $\boldsymbol{X}^n \in \mathbb{R}^{H \times W \times 3}$ denotes the ground truth point map unprojected from depth map, $\boldsymbol{M}^n \in \mathbb{R}^{H \times W}$ is the valid depth mask. To be noted, we also scale the ground truth depths when scaling the cameras.

**Scene Scale Ambiguity.** Given a set of scene images, we can not infer the actual scale of the scene, since we can obtain identical rendered images when scaling the scene and cameras simultaneously, *i.e.*, the scene scale ambiguity problem. For object-centric reconstruction, we have fixed the object center at $(0, 0, 2)$ in the reference frame, so there is no scale ambiguity problem. But for scene images, it is hard to define a canonical center and it is reasonable for our model to reconstruct the scene in arbitrary scale. Besides, the training data is a mixture of multiple datasets that vary greatly in scale, which may make the model confused on scale prediction. Then a question arises, how can we align the scale of the model's prediction with the scale of the training cameras so that we can render target views for supervision?

As mentioned above, we scale the camera locations in a training batch using $s = 1/d$ ($d$ is defined in Equation 9), which can bound the scene into a range reasonable for the model to predict. However, the scale factor is related to the choice of valid points, leading to different scene scales with a minimal difference in selected points. The model would get confused if we train it to reconstruct exactly in this scale since it has no idea which points we utilized to compute this scale. Inspired by DUSt3R (Wang et al., 2024b), we choose to rescale the reconstructed Gaussians before rendering them to deal with scene ambiguity. We first compute the scale factor $\hat{s} = 1/\hat{d}$ using Equation 9 with predicted Gaussian positions $\hat{\boldsymbol{X}}^n$ and the same mask $\boldsymbol{M}^n$. Then we adjust the *position* and *scale* of each Gaussian as $\hat{\boldsymbol{\mu}}_k = \hat{s} \cdot \boldsymbol{\mu}_k, \hat{\boldsymbol{s}}_k = \hat{s} \cdot \boldsymbol{s}_k$. With this operation, we do not care about the absolute scale of the reconstructed scene, but we expect it to align with the ground truth scene after scaling them using the factors $\hat{s}$ and $s$ respectively. In the pre-training stage, we compute $\mathcal{L}_{\text{pos}}$ using the scaled predicted positions and ground truth positions too.

## A.2 Additional Experiments

### A.2.1 Comparison with PF-LRM

PF-LRM is a highly-relevant work to our FreeSplatter-O model. We make a comparison with it on its GSO and OmniObject3D evaluation datasets, both quantitatively and qualitatively.

**Quantitative comparison.** We show the quantitative results in Table 4 and Table 5. On both sparse-view reconstruction and pose estimation tasks, our model achieves lower metrics than PF-LRM on GSO dataset and higher metrics on Omni3D dataset. However, we point out that the images in PF-LRM's GSO evaluation set are rendered with the same rendering engine and hyper-parameters (e.g., light intensity) as PF-LRM's training dataset, which is beneficial for PF-LRM's evaluaton metrics due to the smaller gap between its training and evaluation datasets. On the contrary, the images in the Omni3D evaluation set are not manually rendered but are from the original dataset itself. Therefore, we argue that Omni3D is a more fair benchmark for comparison, and our model consistently outperforms PF-LRM on this benchmark.

**Qualitative comparison.** We also show qualitative comparison results on both datasets in Figure 7, which demonstrate that our FreeSplatter synthesizes significantly better visual details than PF-LRM.

### A.2.2 More Results on Image-to-3D Generation

In this section, we provide **a large set** of visualizations on image-to-3D generation by combining our FreeSplatter-O with different multi-view diffusion models.

Table 4: **Quantitative comparison with PF-LRM on sparse-view reconstruction.**

| Method | GSO | | | OmniObject3D | | |
|---|---|---|---|---|---|---|
| | PSNR ↑ | SSIM ↑ | LPIPS ↓ | PSNR ↑ | SSIM ↑ | LPIPS ↓ |
| | Evaluate renderings at G.T. novel-view poses | | | | | |
| PF-LRM | 25.08 | 0.877 | 0.095 | 21.77 | 0.866 | 0.097 |
| FreeSplatter-O | 23.54 | 0.864 | 0.100 | 22.83 | 0.876 | 0.088 |
| | Evaluate renderings at predicted input poses | | | | | |
| PF-LRM | 27.10 | 0.905 | 0.065 | 25.86 | 0.901 | 0.062 |
| FreeSplatter-O | 25.50 | 0.897 | 0.076 | 26.49 | 0.926 | 0.050 |

Table 5: **Quantitative comparison with PF-LRM on pose estimation.**

| Method | GSO | | | | OmniObject3D | | | |
|---|---|---|---|---|---|---|---|---|
| | RRE ↓ | RRA@15° ↑ | RRA@30° ↑ | TE ↓ | RRE ↓ | RRA@15° ↑ | RRA@30° ↑ | TE ↓ |
| PF-LRM | 3.99 | 0.956 | 0.976 | 0.041 | 8.013 | 0.889 | 0.954 | 0.089 |
| FreeSplatter-O | 8.96 | 0.909 | 0.936 | 0.090 | 3.446 | 0.982 | 0.996 | 0.039 |

**Comparison with pose-dependent LRMs.** In Figure 8 and Figure 9, we compare FreeSplatter's image-to-3D generation results with pose-dependent LRMs, *i.e.*, InstantMesh and LGM, using Zero123++ v1.2 (Shi et al., 2023) and a recent model Hunyuan3D Std (Yang et al., 2024b) as the multi-view generator, respectively. For each input image, we fix the random seed to generate multi-view images, and visualize the novel view synthesis results of each reconstruction model. From the results, we can observe that FreeSplatter generates significantly more clear views and preserves the geometry and texture details better than other baselines.

**Using input image to enhance the results.** In Figure 10, we show another interesting use case of FreeSplatter, i.e., using the input image to enhance the image-to-3D generation results. For multi-view diffusion models like Zero123++ v1.1 (Shi et al., 2023), it generates 6 views from an input image at pre-defined poses, but **the pose of the input image is unknown** (its azimuth is known to be 0° but elevation is unknown). In this case, classical pose-dependent LRMs cannot leverage the input image for reconstruction, but FreeSplatter is able to do this! As Figure 10 shows, using the input image apart from generated views can significantly enhance the reconstruction results in many cases, especially for contents that Zero123++ struggles to generate, *e.g.*, human faces. Compared to generated views, the input image is often more high-quality and contains richer visual details. The pose-free nature of FreeSplatter make it capable of exploiting these precious information in the reconstruction process.

**Recovering the predefined camera poses of multi-view diffusion models.** In Figure 11, we show that FreeSplatter can faithfully recover the pre-defined camera poses of existing multi-view diffusion models from its reconstructed Gaussian maps. This demonstrates its robustness to generated multi-view images which may contain inconsistent contents.

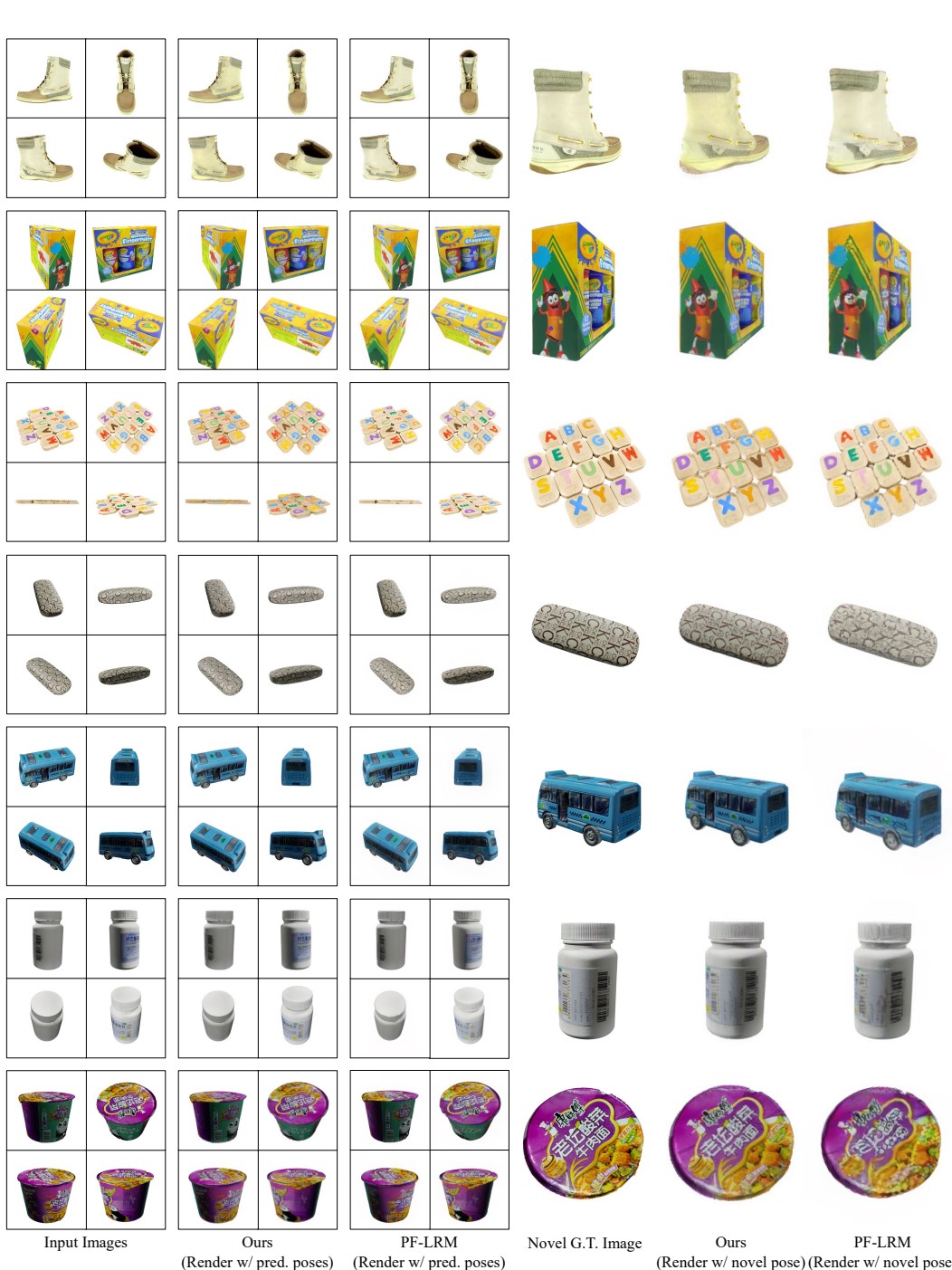

Input Images    Ours (Render w/ pred. poses)    PF-LRM (Render w/ pred. poses)    Novel G.T. Image    Ours (Render w/ novel pose)    PF-LRM (Render w/ novel pose)

Figure 7: **Comparison with PF-LRM on its evaluation datasets.** The test samples in the first 3 rows are from the GSO evaluation set of PF-LRM, while the samples in the last 4 rows are from the OmniObject3D evaluation set of PF-LRM. Our FreeSplatter-O synthesizes significantly better visual details than PF-LRM.

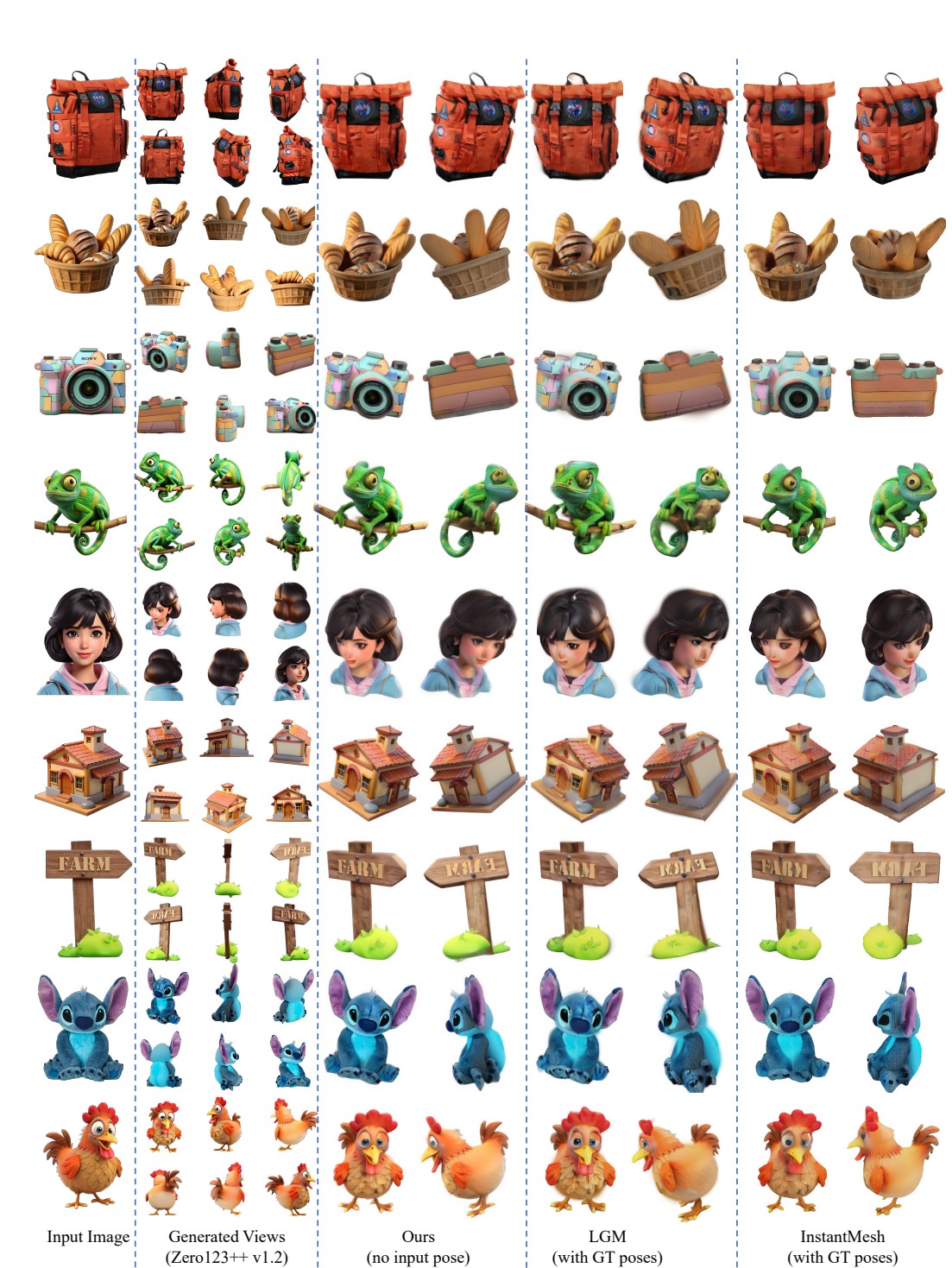

Figure 8: **Comparison on image-to-3D generation with Zero123++ v1.2 (Shi et al., 2023).**

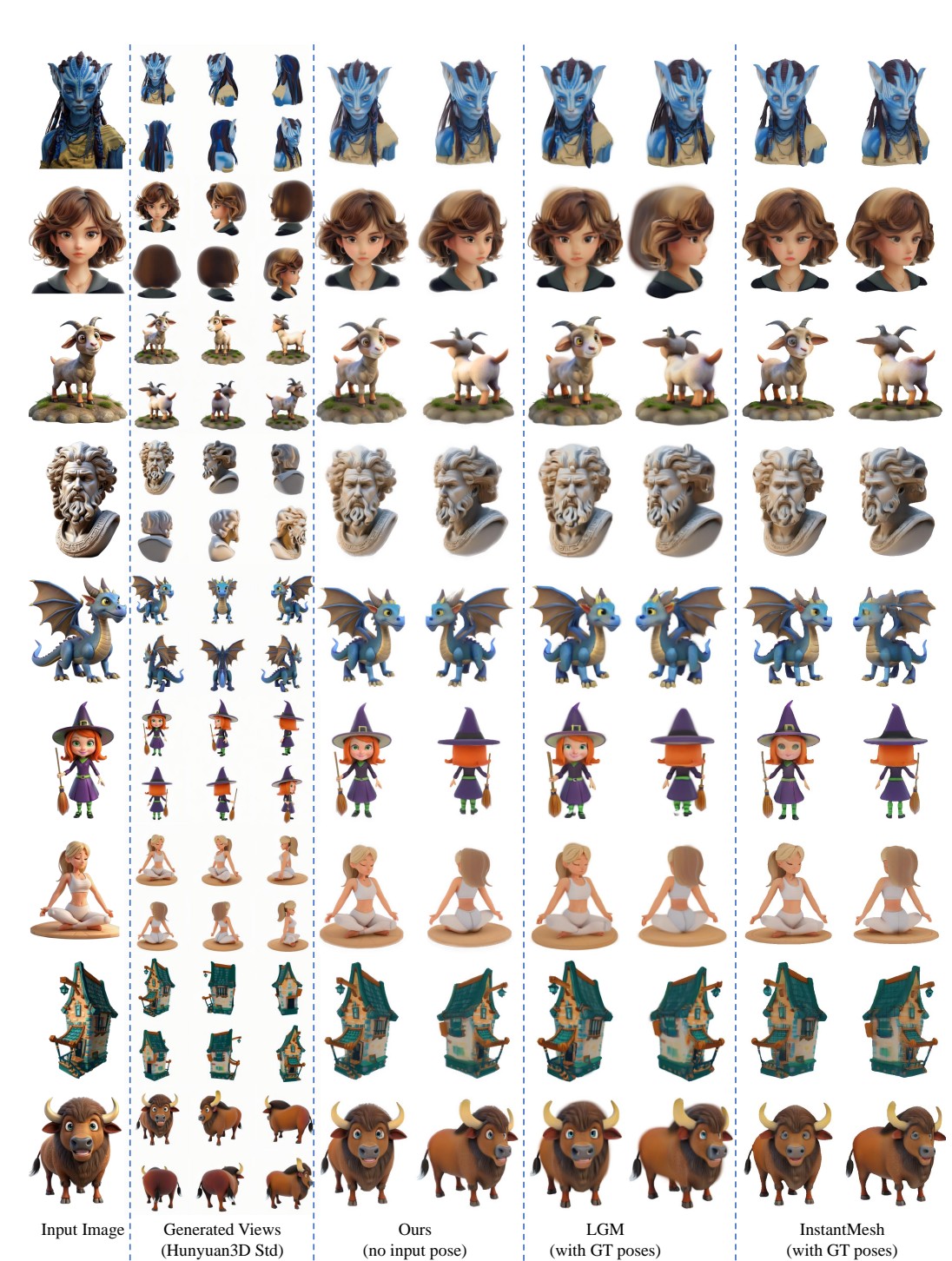

Figure 9: **Comparison on image-to-3D generation with Hunyuan3D Std** (Yang et al., 2024b).

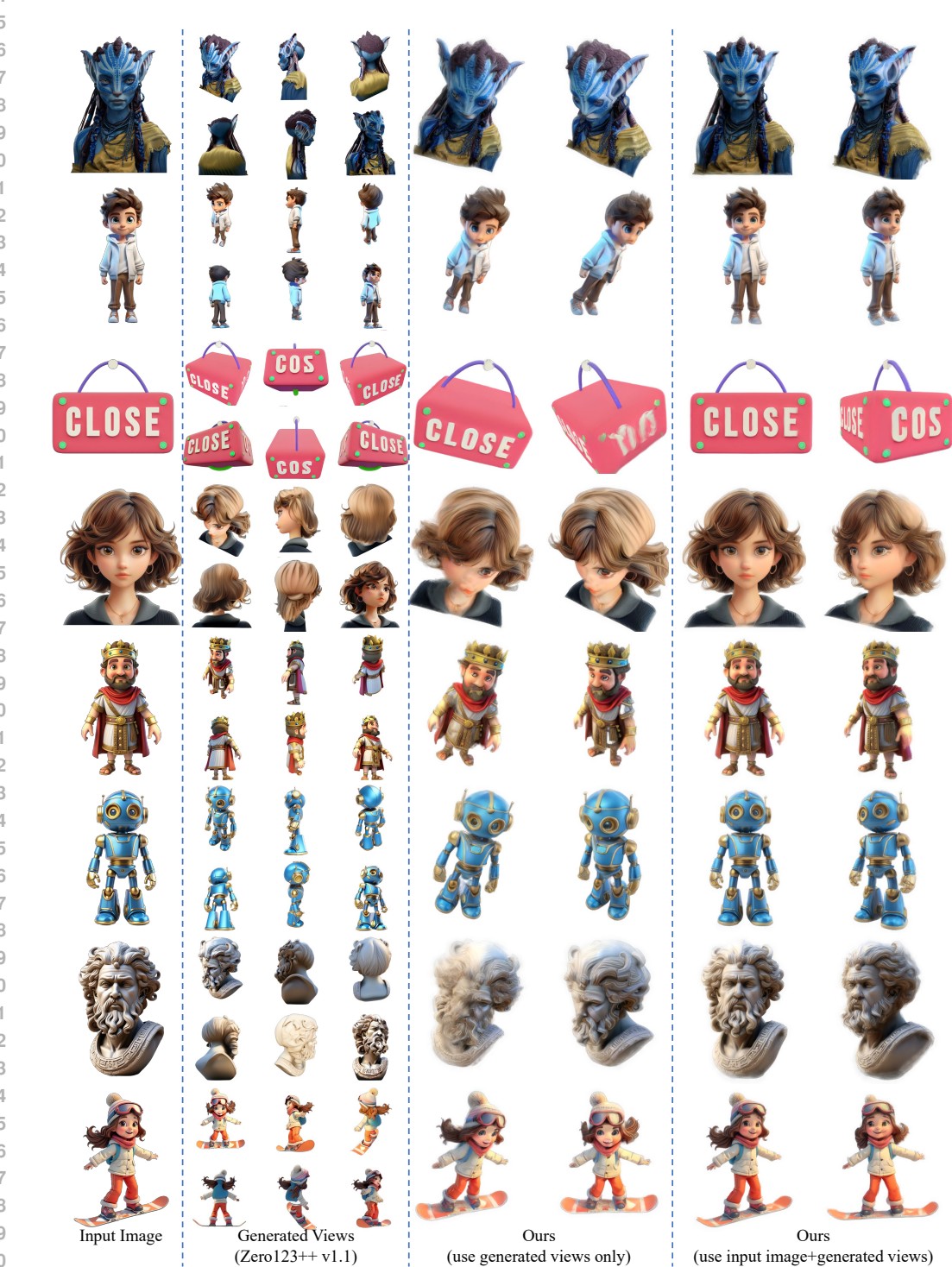

Figure 10: **Use input image to enhance the image-to-3D generation results with Zero123++ v1.1** (Shi et al., 2023). The input image is often more high-quality and contains richer visual details than the generated views, but its camera pose is unknown, making it impossible for pose-dependent LRMs to leverage it. The capability of using the input image alongside generated views of our FreeSplatter is particularly valuable for challenging content like human faces, where Zero123++ often struggles to generate.

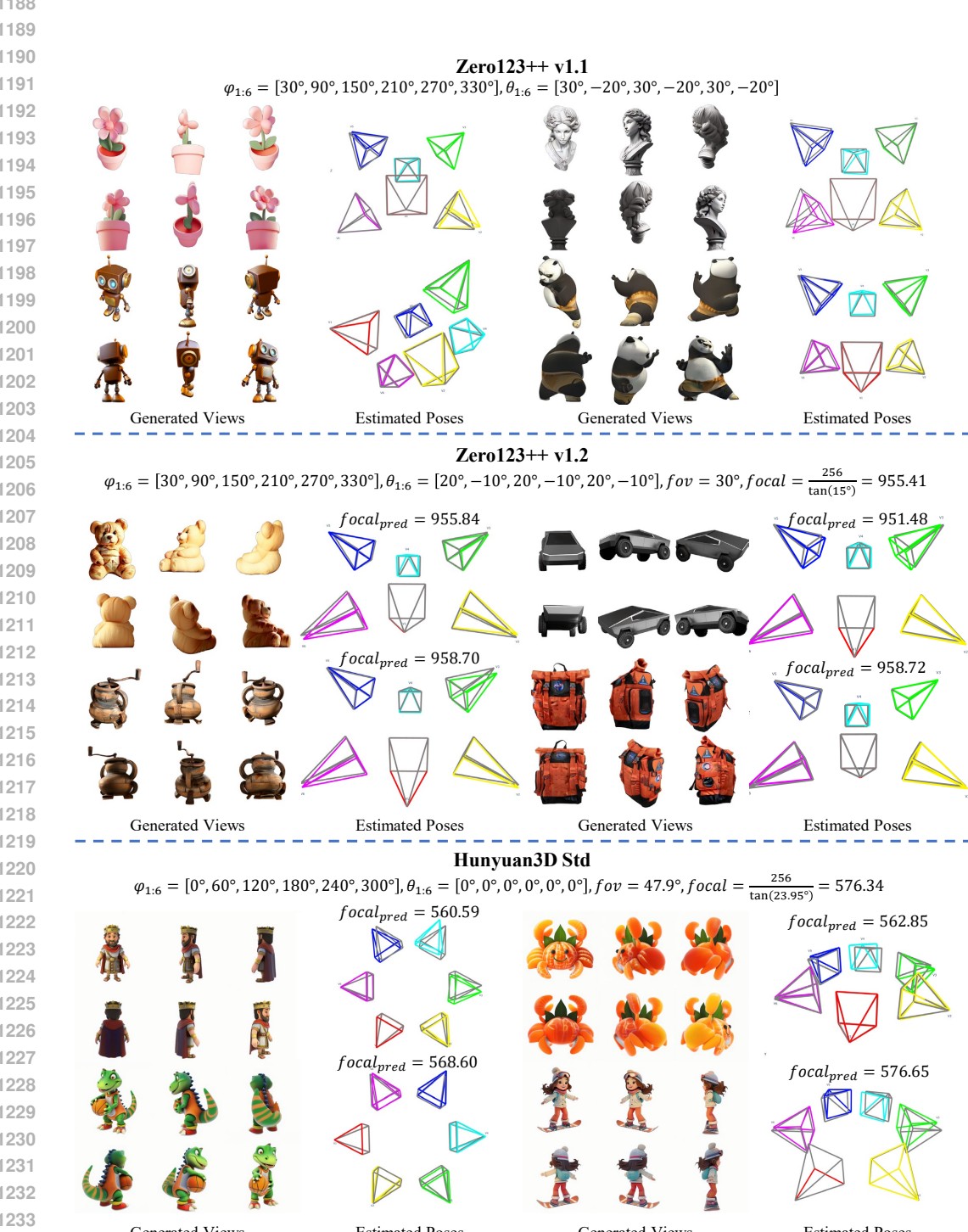

Figure 11: **Our FreeSplatter can faithfully recover the pre-defined camera poses of existing multi-view diffusion models.** We use gray pyramids to visualize the ground truth pre-defined camera poses of the diffusion models, and colorful pyramids to visualize the estimated poses. $\varphi$ and $\theta$ denote the pre-defined azimuth and elevation angles, respectively. Since Zero123++ v1.2 and Hunyuan3D Std generate images at pre-defined fixed Field-of-View (fov), we mark their pre-defined fov angles and corresponding focal lengths (in pixel units) too.

### A.2.3 CROSS-DATASET GENERALIZATION

**FreeSplatter-O generalizes to other object datasets.** To demonstrate the cross-dataset generalization ability of our FreeSplatter-O, we visualize the object-centric reconstruction results on OmniObject3D and ABO (Collins et al., 2022) datasets in Figure 12. Both datasets are outside the training scope of FreeSplatter-O. We can observe that FreeSplatter-O generates very high-quality renderings.

**FreeSplatter-S generalizes to RealEstate10K.** Although our FreeSplatter-S model was not trained on RealEstate10K (Zhou et al., 2018), we directly evaluate its zero-shot view synthesis capability on this dataset. We visualize the qualitative results in Figure 13, showing that our model can faithfully reconstruct the input views at the estimated input poses, while the rendered novel views align well with the ground truth. We also report the quantitative results of both sparse-view reconstruction and pose-estimation tasks in Table 6 and Table 7, respectively. In summary, pixelSplat and MVS-plat achieve better metrics on the sparse-view reconstruction task, which is foreseeable since they utilize RealEstate10K as training dataset and ground truth camera poses as input, while our model is not trained on this dataset and leverages no camera information. However, our model significantly outperforms another pose-free baseline Splatt3R which is built upon MASt3R trained on a larger dataset, demonstrating the effectiveness of our framework. For camera pose estimation on RealEstate10K, MASt3R achieves the best Relative Rotation Error (RRE) while our FreeSplatter-S performs the best in all other metrics.

Table 6: **Quantitative sparse-view reconstruction results on RealEstate10K.**

| Method | PSNR ↑ | SSIM ↑ | LPIPS ↓ |
|---|---|---|---|
| pixelSplat (w/ GT poses) | 24.469 | 0.829 | 0.224 |
| MVSplat (w/ GT poses) | 20.033 | 0.789 | 0.280 |
| Splatt3R (no pose) | 16.634 | 0.604 | 0.422 |
| FreeSplatter-S (no pose) | 18.851 | 0.659 | 0.369 |

Table 7: **Quantitative pose estimation results on RealEstate10K.**

| Method | RRE ↓ | RRA@15° ↑ | RRA@30° ↑ | TE ↓ |
|---|---|---|---|---|
| PoseDiffsion | 14.387 | 0.732 | 0.780 | 0.466 |
| RayDiffsion | 12.023 | 0.767 | 0.814 | 0.439 |
| RoMa | 5.663 | 0.918 | 0.947 | 0.402 |
| MASt3R | 2.341 | 0.972 | 0.994 | 0.374 |
| FreeSplatter-S | 3.513 | 0.982 | 0.995 | 0.293 |

**FreeSplatter-S generalizes to various other datasets.** We demonstrate FreeSplatter-S's broad generalization capabilities through extensive evaluation across diverse datasets in Figure 14. Our results span scanned objects (DTU dataset), large-scale outdoor scenes (Tanks & Temples), multi-view object collections (MVImgNet), and AI-generated synthetic contents (Sora-generated videos[2]). These results validate our framework's robustness across varying scene types and capture conditions.

---

[2] https://openai.com/index/sora/

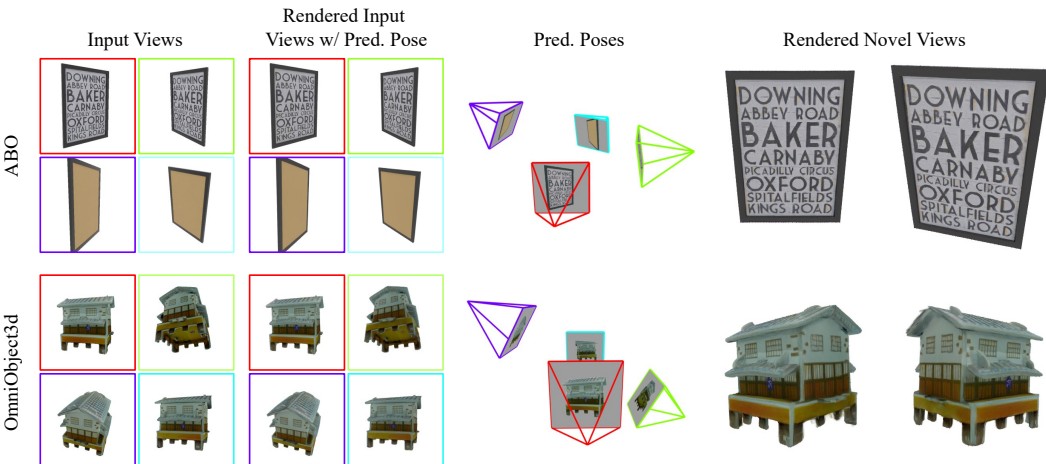

Figure 12: **Zero-shot pose-free reconstruction results on ABO and OmniObject3D.** Both datasets are unseen for FreeSplatter-O. Our model can faithfully estimate the input camera poses and render high-fidelity novel views.

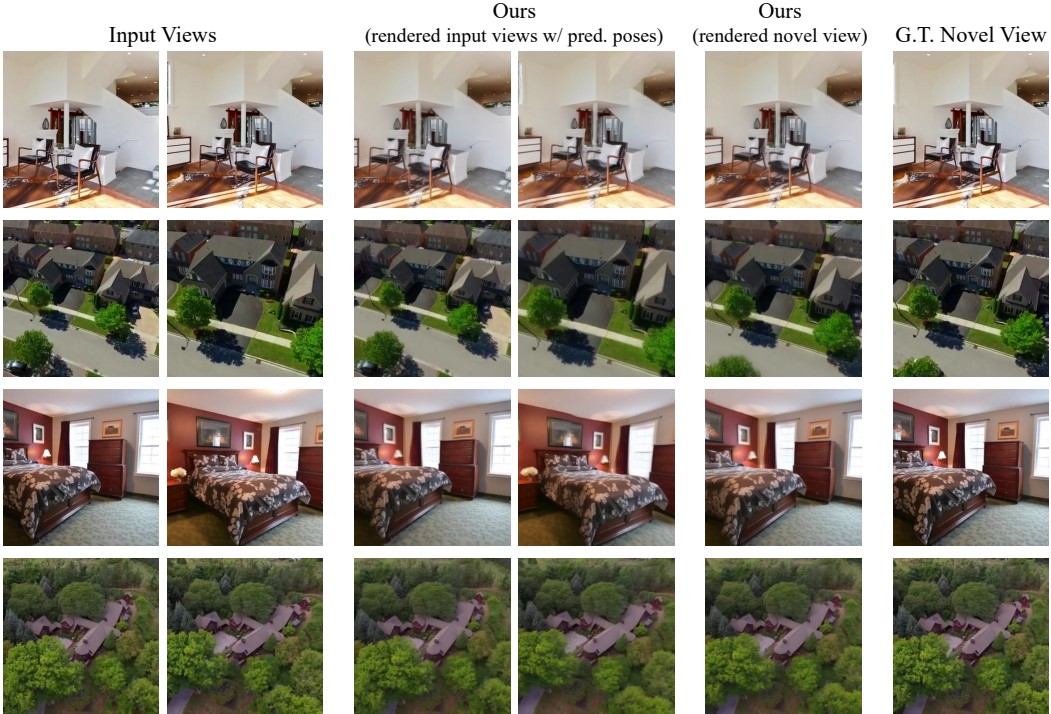

Figure 13: **Zero-shot pose-free reconstruction and view synthesis results on RealEstate10K.** Our FreeSplatter-S model was not trained on RealEstate10K, we directly utilize it for zero-shot view synthesis on this dataset. We can observe that our model can faithfully reconstruct the input views at the estimated input poses, and the rendered novel views align well with the ground truth.

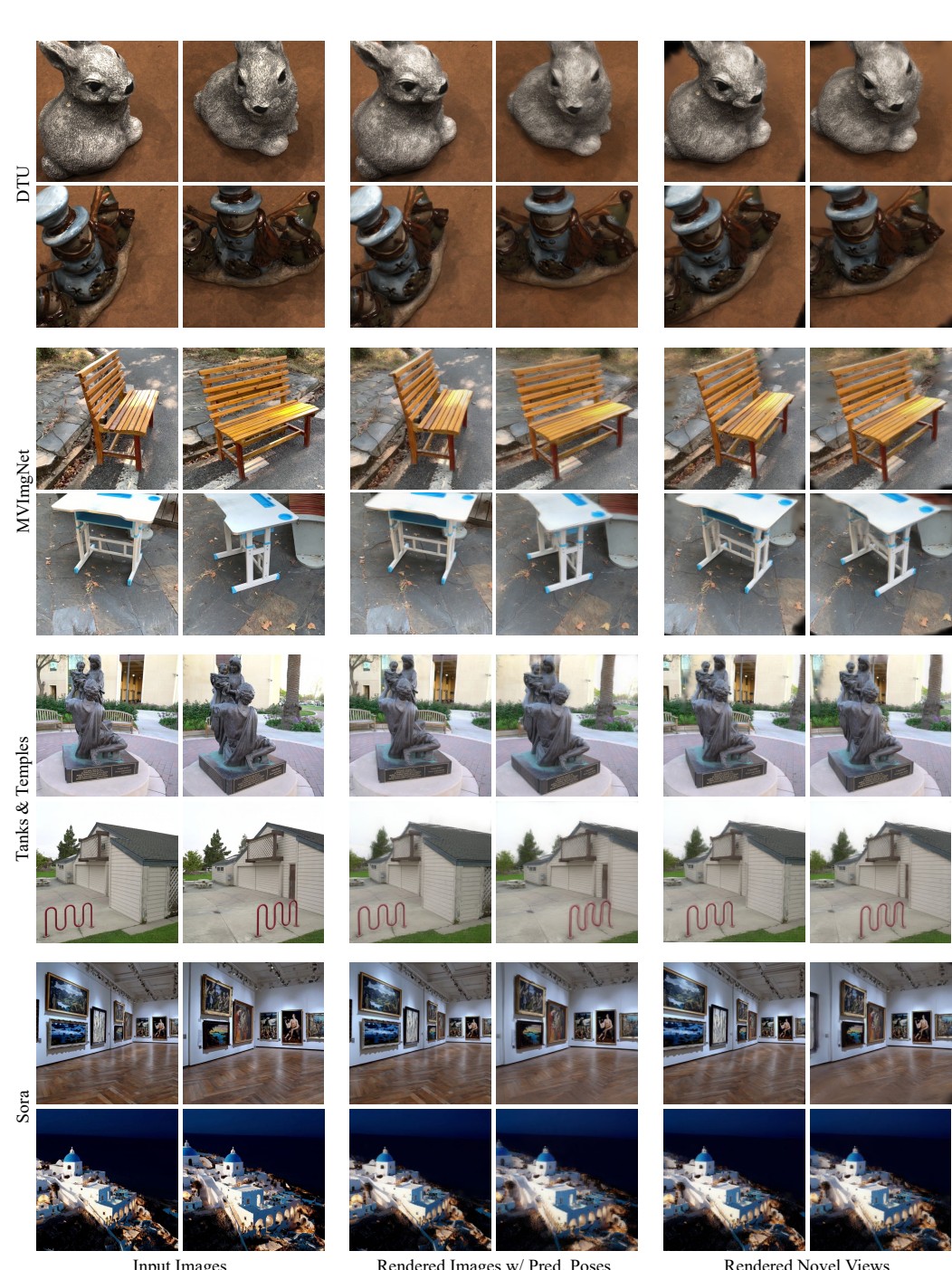

Figure 14: **Zero-shot generalization of FreeSplatter-S on various datasets.** We show 2 examples for DTU, MVImgNet, Tanks & Temples and Sora-generated videos, respectively.

## A.3 ABLATION STUDIES

**Model architecture.** Our FreeSplatter model is a 24-layer transformer with a patch size of 8. In our initial experiments of this project, we have trained object-level reconstruction models with different layers and patch size. We compare these models with our final model to evaluate the influence of model architecture on the performance. Specifically, we evaluate on the Google Scanned Objects dataset and report the results in Table 8. The results demonstrate that, using more layers and smaller patch size can consistently improve the model's performance. This is reasonable since more transformer layers leads to more parameters, while smaller patch size leads to less memory usage (less image tokens for attention computation) but higher information compression ratio.

Table 8: **Ablation study on model architectures.** The results are evaluated on GSO dataset. $L$ and $P$ denote number of transformer layers and patch size, respectively.

| Architecture | PSNR ↑ | SSIM ↑ | LPIPS ↓ |
|---|---|---|---|
| $L = 16, P = 16$ | 25.417 | 0.896 | 0.088 |
| $L = 16, P = 8$ | 28.945 | 0.934 | 0.064 |
| $L = 24, P = 16$ | 28.622 | 0.927 | 0.063 |
| $L = 24, P = 8$ | 30.443 | 0.945 | 0.055 |

**View embedding addition.** We also evaluate the effectiveness of adding view embeddings to images tokens. In FreeSplatter, we add the multi-view image tokens with a reference view embedding $\mathbf{e}_{\text{ref}}$ or a source view embedding $\mathbf{e}_{\text{src}}$ before feeding them into the transformer to make the model identify the reference view, so that it can reconstruct Gaussians in the reference view's camera frame. We present the results in Figure 17 of the revised paper.

Specifically, for 4 input views $V_i$ ($i = 1, 2, 3, 4$), we try different combinations of $\mathbf{e}_{\text{ref}}$ and $\mathbf{e}_{\text{src}}$ when adding them to the image tokens, and then render the reconstructed Gaussians with an "identity camera pose" $C = [[1, 0, 0, 0], [0, 1, 0, 0], [0, 0, 1, 0], [0, 0, 0, 1]]$ to see the results. We can observe that, when adding the $j$-th view's tokens with $\mathbf{e}_{\text{ref}}$ and other views' tokens with $\mathbf{e}_{\text{src}}$, the rendered image is exactly the $j$-th view. This means that the model successfully identify the $j$-th view as the reference view and reconstruct Gaussians in its camera frame. In comparison, all other view embedding combinations leads to degraded reconstruction.

**Qualitative comparison of adding pixel-alignment loss.** Apart from the quantitative results in Table 3, we visualize the influence of pixel-alignment loss on sparse-view reconstruction results in Figure 15 to better demonstrate its effectiveness. The results show that the model trained without pixel-alignment loss leads to blurry novel view renderings, while the model trained with pixel-alignment loss demonstrates significantly better visual details.

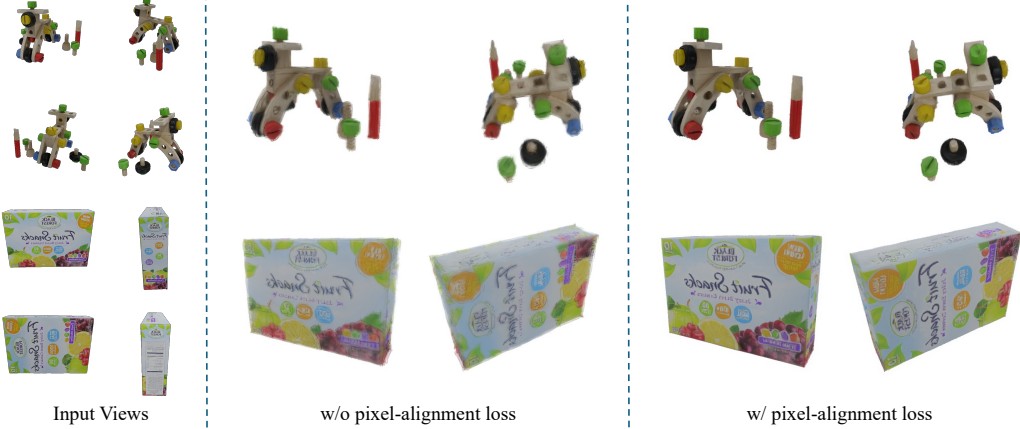

| Input Views | w/o pixel-alignment loss | w/ pixel-alignment loss |

Figure 15: **Ablation on pixel-alignment loss.** We show two samples from the GSO dataset.

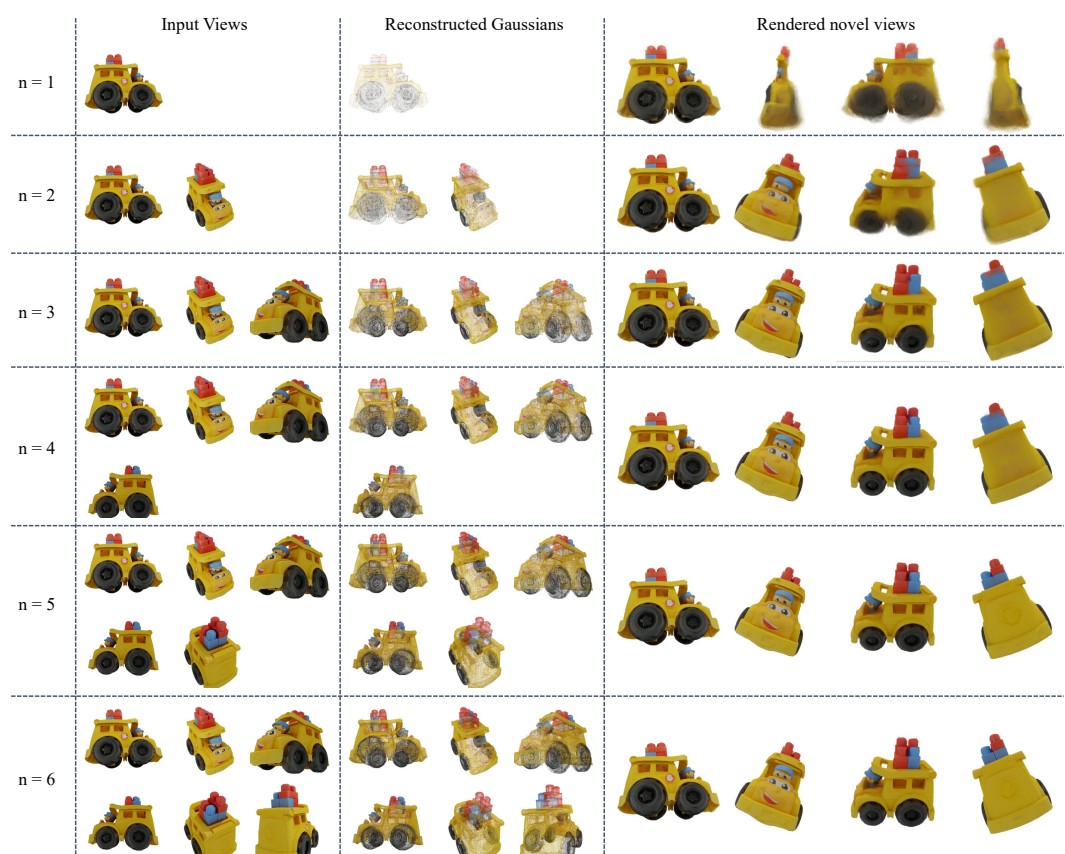

Figure 16: **Illustration on the influence of input view number.** We show the visual comparison of FreeSplatter-O results with varying numbers of input views ($n = 1 - 6$). From left to right: input views, reconstructed Gaussians, and rendered target views at 4 fixed viewpoints. Additional input views increase Gaussian density and improve previously uncovered regions, with diminishing returns beyond n=4 when object coverage becomes sufficient.

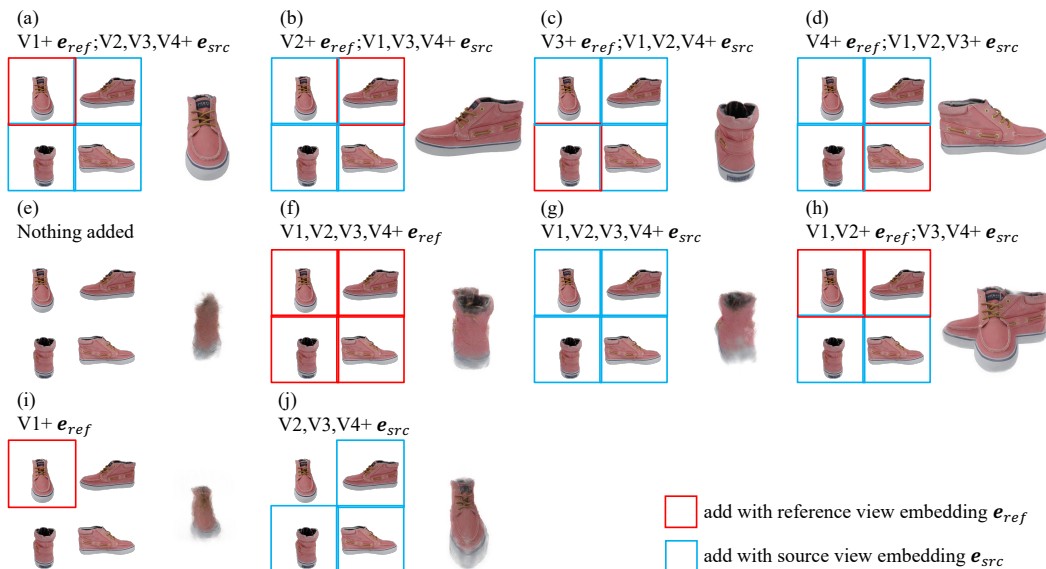

Figure 17: **Ablation study on view embedding addition.** Red/blue boxes indicate views added with reference/source view embeddings respectively. For each case, we visualize the image rendered with identity camera (*i.e., reference pose*) on the right.

