# OpenReview forum: "FreeSplatter: Pose-free Gaussian Splatting for Sparse-view 3D Reconstruction"
_ICLR.cc/2025/Conference — Submitted to ICLR 2025_

### Official Review · Reviewer_sB8Q · 2024-10-31

**Soundness:** 3
**Presentation:** 3
**Contribution:** 2
**Rating:** 5
**Confidence:** 5

**Summary:**

The paper presents FreeSplatter, a scalable feed-forward framework for reconstructing 3D Gaussian representations from uncalibrated sparse-view images while simultaneously recovering camera parameters. To cover both object-centric and scene-level settings, the authors train two variant models on a large amount of 3D dataset. The proposed method addresses challenges associated with fast pose-free sparse-view reconstruction and accurate camera pose estimation.

**Strengths:**

1. FreeSplatter addresses the problem of pose-free Gaussian Splatting for sparse-view reconstruction in a feed-forward and scalable manner.
2. The idea of integrating camera parameter recovery into the feed-forward framework with 3DGS reconstruction showcases is insightful.
3. Using a streamlined transformer architecture is straightforward and useful as the experiments suggest.
4. The method’s scalability is advantageous, making it suitable for large-scale real-world applications.

**Weaknesses:**

1. The pipeline contribution is limited as the architecture design is quite similar to GS-LRM [1] without specific designs for accurate camera pose estimation. The camera pose is estimated after generating Gaussian Maps which is independent in the feed-forward framework and more likely to a post-processing stage with the PnP algorithm. This makes the paper paradigm like just A (GS-LRM architecture) plus B (PnP algorithm).
2. A more elegant design may incorporate camera pose estimation into the feed-forward framework (e.g., generating image tokens with camera tokens simultaneously).
3. The author cannot address the occluded areas in a scene-level setting and the strategy in object-centric is quite similar to the Splatter Image [2]. This also hurt the contribution of the paper. Maybe the insight in Flash3D [3] can be incorporated into the feed-forward architecture.



[1] Zhang K, Bi S, Tan H, et al. GS-LRM: Large reconstruction model for 3d gaussian splatting; ECCV 2024.

[2] Szymanowicz S, Rupprecht C, Vedaldi A. Splatter image: Ultra-fast single-view 3d reconstruction; CVPR 2024.

[3] Szymanowicz S, Insafutdinov E, Zheng C, et al. Flash3D: Feed-Forward Generalisable 3D Scene Reconstruction from a Single Image; arXiv 2024.

**Questions:**

1. How is the generalizability of the FreeSplatter? As the author mentioned the pipeline is scalable and can be trained on large data, I want to see more discussion about the generalizability of various datasets (e.g., the difference of angles of input views or indoor scene generalize to outdoor scene).
2. Have you considered to train object-level and scene-level data in a unified framework? What is the key challenge to realize the goal?
3. As you mentioned the scalability of the FreeSplatter, why do you only use two input views to train the framework in a scene-level setting (just same as GS-LRM)? How is the performance under more views?

---

> ### Author Response · Authors · 2024-11-25
> **Response to Reviewer sB8Q (1/2)**
>
> > The pipeline contribution is limited as the architecture design is quite similar to GS-LRM without specific designs for accurate camera pose estimation.
>
> In this work, we focus on liberating the sparse-view reconstruction task from camera poses rather than making architectural innovation. In fact, the architecture of GS-LRM itself is a simple Vision Transformer with a Gaussian-parameter-prediction head, which is a stack of self-attention layers. We demonstrate that such an elegant architecture can effectively handle pose-free reconstruction through direct image-to-Gaussian mapping, enabling both high-quality 3D modeling and efficient pose estimation.
>
> > The camera pose is estimated after generating Gaussian Maps which is independent in the feed-forward framework and more likely to a post-processing stage with the PnP algorithm. This makes the paper paradigm like just A (GS-LRM architecture) plus B (PnP algorithm).
>
> We respectfully emphasize that our work addresses **pose-free** sparse-view reconstruction without camera poses or intrinsics as input, presenting significantly greater challenges than GS-LRM's pose-dependent setting. While GS-LRM predicts per-pixel depth values and Gaussian attributes with straightforward rendering loss supervision, directly adopting this approach in our pose-free setting leads to **non-convergence**. To accomplish the pose-free reconstruction pipeline, we have made substantial efforts:
>
> - In a pose-free setting, the camera poses and intrinsics are unknown to the model, thus it is infeasible to back-project the 3D positions of Gaussians by predicting depth values only. Therefore, we parameterize the Gaussian positions with 3D coordinates and make the model learn to predict them.
> - In the training stage of GS-LRM, the Gaussian positions are restricted on the camera rays, significantly reducing the difficulty of predicting correct Gaussian positions with only rendering supervision. However, the Gaussian positions predicted by our model can move freely in the 3D space, so it is very challenging to guide the Gaussian to its correct position with only the rendering loss. To address this issue, we propose a two-stage training strategy by initializing the Gaussian positions with depth guidance first, and then refine them with rendering loss, which demonstrates to be very effective.
> - To furthre improve the 3D reconstruction quality and pose estimation accuracy, we propose to adopt a pixel-alignment loss for supervision in the second stage. This loss enforces the Gaussian positions to lie on ground truth camera rays and prevents them from moving everywhere, significantly enhancing the rendering quality. Meanwhile, this loss also ensures the pose-estimation accuracy at test time, since the PnP solver used for pose estimation minimizes the point-pixel projection errors, which can also be minimized by enforcing the pixel-alignemnt property of Gaussian positions.
>
> > A more elegant design may incorporate camera pose estimation into the feed-forward framework (e.g., generating image tokens with camera tokens simultaneously).
>
> Generating image tokens with camera tokens simultaneously means that the model needs to learn to regress the accurate camera poses directly. We argue that this presents fundamental challenges compared to our Gaussian-map-based approach. In this framework, determinating the Gaussian positions relies on the rigid-body transformations defined by the regressed camera poses, where minor pose regression errors will be propagated and amplified in the transformed Gaussians, degrading the reconstruction quality. In comparison, our multi-view Gaussian map approach provides an over-parameterized pose representation, effectively distributing the risk of model prediction errors across numerous points and enhancing model robustness.
>
> > The author cannot address the occluded areas in a scene-level setting and the strategy in object-centric is quite similar to the Splatter Image.
>
> While we acknowledge the importance of occlusion handling, our primary contribution lies in developing an elegant yet effective pose-free reconstruction framework that unifies both object-centric and scene-level tasks. Although our approach to enforcing pixel-alignment in foreground regions proves effective for handling invisible areas in object-centric scenarios, we do not position this as one of our primary contributions. We appreciate the reviewer's reference to Flash3D's layered Gaussian approach and will consider integrate its spirit into our framework as a promising direction for future research addressing occlusion challenges.

---

> ### Author Response · Authors · 2024-11-25
> **Response to Reviewer sB8Q (2/2)**
>
> > I want to see more discussion about the generalizability of various datasets.
>
> We demonstrate FreeSplatter's broad generalization capabilities through extensive evaluation across diverse datasets in **Figure 14** of the revised paper. Our results span:
> - Indoor environments (DTU dataset)
> - Large-scale outdoor scenes (Tanks & Temples)
> - Multi-view collections (MVImgNet)
> - Synthetic content (Sora-generated videos)
>
> These results validate our framework's robustness across varying scene types and capture conditions.
>
> > Have you considered to train object-level and scene-level data in a unified framework? What is the key challenge to realize the goal?
>
> We have indeed considered training a unified model for both tasks, but it poses the following challenges:
> 1. **Scale Discrepancy.** Object-level datasets are typically rendered from synthetic 3D datasets (e.g., Objaverse), the objects are often normalized a unit cube before rendering, leading to bounded and consistent object scales. However, scene-level datasets exhibit significant scale variations and standardized scale normalization, challenging the model's scale inference capabilities.
> 2. **Camera Distribution Divergence.** In object-centric scenarios, the common practice is to normalize and translate the object center to the origin and sample cameras uniformly around the object which face towards the origin. The models trained for object-level reconstruction can leverage this centered object prior. However, the cameras in scene-level scenarios are much more complex, which can face arbitrary camera orientations and have no defined "scene center".
> 3. **Region of Interest Variations.** Object-level reconstruction focuses on the foreground object, while the background is assumed to be uniform-colored (e.g., pure white) and are neglected in the reconstruction process. However, all visible areas are relevant in scene-level reconstruction and there is no clear foreground/background distinction. This leads to challenges in unified attention allocation when training a model for object-centric and scene-level reconstruction jointly.
>
> > Why do you only use two input views to train the framework in a scene-level setting (just same as GS-LRM)? How is the performance under more views?
>
> Our current implementation with two-view training stems from our focus on fair comparison with state-of-the-art pose-free reconstruction frameworks, particularly DUSt3R and MUSt3R. To ensure direct comparability, we train FreeSplatter-S on a subset of their training datasets and utilize the pre-processed paired training views provided by DUSt3R.
>
> While our model architecture is fundamentally view-count agnostic and makes no assumptions about input view quantity, practical considerations currently constrain our implementation. The existing model, optimized for two-view input, may exhibit artifacts when processing additional views due to increased Gaussian accumulation. This limitation arises from the current training protocol rather than architectural constraints.
>
> Looking forward, we envision expanding our framework's capabilities through comprehensive dataset reprocessing and model scaling. Our architecture's inherent flexibility enables natural extension to multiple views, and we anticipate significant performance improvements through increased view utilization. This development aligns with our broader goal of creating a more versatile reconstruction system while maintaining the current framework's effectiveness in pose-free scenarios.

---

> ### Author Response · Authors · 2024-11-27
> **Kind reminder for discussion**
>
> Dear Reviewer sB8Q,
>
> As we approach the end of the discussion period, we wish to kindly remind you of our responses and ensure we have adequately addressed your concerns. We have carefully revised our manuscript based on your valuable feedback, with modifications highlighted in blue for easy reference.
>
> If you have any remaining questions or require additional clarification, we would be glad to provide further information.
>
> Best regards, FreeSplatter Authors

---

> > ### Comment · Reviewer_sB8Q · 2024-11-30
> >
> > Thanks for your detailed response. After carefully considering your responses and the comments of the other reviewers. I keep my original rating, primarily due to the contribution and novelty of the paper.

---

### Official Review · Reviewer_nq6R · 2024-11-03

**Soundness:** 1
**Presentation:** 3
**Contribution:** 3
**Rating:** 5
**Confidence:** 5

**Summary:**

This paper proposes a feed-forward method, FreeSplatter, for reconstructing 3D Gaussian Splatting from pose-free sparse images. The proposed method concatenates input images, which are then processed by a Vision Transformer network that outputs 3D Gaussians in a reference input view. The predicted Gaussians can be used for novel view synthesis and to estimate camera pose through PnP. FreeSplatter is trained on both object-level and scene-level datasets, and the results demonstrate SOTA performance across on both object-level and scene-level datasets.

**Strengths:**

- The proposed method demonstrates impressive results for novel view synthesis with unposed inputs.
- The pose-free pipeline is promising and produces strong results.
- The network architecture of the proposed method is simple but effetive.
- The paper is well-written and easy to understand.

**Weaknesses:**

- Lack of fair quantitative comparisons with methods such as pixelSplat and MVSplat. Although Figure 8 presents qualitative results on RE10K, quantitative results are absent. Even though the proposed method requires ground-truth depth information for training, it can still be trained on RE10K by initially training on ScanNet++ and then fine-tuning on RE10K.
- Some comparisons with baseline methods may be unfair. For instance, the patch size of the ViT backbone is critical in pixel-aligned Gaussian prediction methods, as it determines the number of Gaussians. The proposed method, following GS-LRM, uses a patch size of 8, whereas pixelSplat and MVSplat use a patch size of 16. How does performance compare if both methods use a patch size of 8?
- In Line 428, the authors claim that the proposed method outperforms pose-dependent methods. However, this claim lacks a fair comparison against pose-dependent methods under similar conditions as discussed before. An ablation study could compare the proposed method to a pose-dependent approach using the same backbone, such as the GS-LRM pipeline.
- Limited comparison with other SOTA pose-estimation methods, such as RoMa [1] for scene-level pose estimation and Cameras as Rays [2] for object-level pose estimation.
- No ablation study on the effectiveness of adding reference and source embeddings.

[1] Edstedt, Johan, et al. "RoMa: Robust dense feature matching." CVPR. 2024.
[2] Zhang, Jason Y., et al. "Cameras as rays: Pose estimation via ray diffusion." ICLR. 2024.

**Questions:**

- As noted in the weaknesses, while the proposed pipeline is promising and demonstrates strong performance, a more rigorous comparison and additional experimental analysis are needed to substantiate the claims and more clearly showcase the method’s effectiveness.
- The results are promising but require significant GPU resources for training, which may make reproduction difficult for the community. Will the code be made available?

---

> ### Author Response · Authors · 2024-11-25
> **Response to Reviewer nq6R (1/2)**
>
> > Although Figure 8 presents qualitative results on RE10K, quantitative results are absent.
>
> We present zero-shot generalization results on RealEstate10K for both sparse-view reconstruction and pose estimation tasks, as comprehensive training on this dataset was constrained by computational resources during the rebuttal period.
>
> **Table 1: Sparse-view Reconstruction Performance on RealEstate10K**
> | **Method** | **PSNR↑** | **SSIM↑** | **LPIPS↓** |
> | --- | --- | --- | --- |
> | pixelSplat (w/ GT poses) | **24.469** | **0.829** | **0.224** |
> | MVSplat (w/ GT poses) | 20.033 | 0.789 | 0.280 |
> | Splatt3R (no pose) | 16.634 | 0.604 | 0.422 |
> | FreeSplatter-S (no pose) | 18.851 | 0.659 | 0.369 |
>
> **Table 2: Camera Pose Estimation Performance on RealEstate10K**
> | **Method** | **RRE↓** | **RRA@$15^\circ$↑** | **RRA@$30^\circ$↑** | **TE↓** |
> | --- | --- | --- | --- | --- |
> | PoseDiffsion | 14.387 | 0.732 | 0.780 | 0.466 |
> | RayDiffsion | 12.023 | 0.767 | 0.814 | 0.439 |
> | RoMa | 5.663 | 0.918 | 0.947 | 0.402 |
> | MASt3R | **2.341** | 0.972 | 0.994 | 0.374 |
> | FreeSplatter-S | 3.513 | **0.982** | **0.995** | **0.293** |
>
> While pose-dependent methods (pixelSplat, MVSplat) achieve superior reconstruction metrics, this is expected given their training on RealEstate10K and access to ground truth poses. Notably, our pose-free approach significantly outperforms Splatt3R, despite the latter being trained on a larger dataset, demonstrating our framework's effectiveness.
>
> In pose estimation, FreeSplatter-S achieves best-in-class performance across most metrics, with only a slight disadvantage in RRE compared to MASt3R.
>
> > Some comparisons with baseline methods may be unfair. How does performance compare if both methods use a patch size of 8?
>
> We contend that our comparisons are methodologically sound, considering:
> 1. Baseline methods benefit from ground truth camera poses;
> 2. We fine-tuned pixelSplat and MVSplat on ScanNet++ for fair comparison.
>
>
> To address the patch size concerns, we evaluated a FreeSplatter-S variant trained in our early experiments with patch size 16 on ScanNet++:
>
> **Table 3: Impact of Patch Size on Performance**
> | **Method** | **PSNR↑** | **SSIM↑** | **LPIPS↓** |
> | --- | --- | --- | --- |
> | pixelSplat (w/ GT poses) | 24.974 | 0.889 | 0.180 |
> | MVSplat (w/ GT poses) | 22.601 | 0.862 | 0.208 |
> | Splatt3R (no pose) | 21.013 | 0.830 | 0.209 |
> | FreeSplatter-S ($P=16$) | 24.145 | 0.878 | 0.152 |
> | FreeSplatter-S ($P=8$) | 25.807 | 0.887 | 0.140 |
>
> Even with larger patch size (P=16), FreeSplatter maintains competitive performance, achieving superior LPIPS and comparable PSNR/SSIM versus pose-dependent alternatives.
>
> > An ablation study could compare the proposed method to a pose-dependent approach using the same backbone, such as the GS-LRM pipeline.
>
> Training LRMs using the same backbone from scratch requires significant GPU resources, which is not practical and affordable for us considering the limited rebuttal period and GPU resources we can mobilize. In fact, the pose-dependent baselines LGM and InstantMesh have 379M and 415M parameters respectively, both are larger than our FreeSplatter model which has 306M parameters. To alleviate the reviewer's concern on the influence of model architecture on the performance, we instead demonstrate our method's efficiency through a reduced-parameter comparison. We evaluated a smaller FreeSplatter-O variant (16 layers, patch size 16, 200M parameters) on the GSO dataset:
>
> **Table 4: Performance Comparison with Reduced Parameters**
> | **Method** | **PSNR↑** | **SSIM↑** | **LPIPS↓** |
> | --- | --- | --- | --- |
> | LGM (w/ GT poses) | 24.463 | 0.891 | 0.093 |
> | InstantMesh (w/ GT poses) | 25.421 | 0.891 | 0.095 |
> | FreeSplatter-O ($L=16, P=16$) | 25.417 | 0.896 | 0.088 |
> | FreeSplatter-O ($L=24, P=8$) | 30.443 | 0.945 | 0.055 |
>
> Despite using approximately half the parameters, our compact model achieves competitive or superior performance across all metrics, demonstrating the advantages of our approach.

---

> ### Author Response · Authors · 2024-11-25
> **Response to Reviewer nq6R (2/2)**
>
> > Limited comparison with other SOTA pose-estimation methods, such as RoMa [1] for scene-level pose estimation and Cameras as Rays [2] for object-level pose estimation.
>
> Following the reviewer's suggestion, we conducted comprehensive comparisons with state-of-the-art pose estimation methods. Note that RayDiffusion (Cameras as Rays) was trained on CO3D dataset, which aligns with our scene-level reconstruction category. Therefore, we compare both RoMa and RayDiffusion with FreeSplatter-S:
>
> **Table 5: Cross-Dataset Pose Estimation Performance**
> | **Method** |  | **ScanNet++** |  |  |  | **CO3Dv2** |  |  |  | **RealEstate10K** |  |  |
> | :--- | :--- | :--- | :--- | :--- | :--- | :--- | :--- | :--- | :--- | :--- | :--- | :--- |
> |  | RRE↓ | RRA@$15^\circ$↑ | RRA@$30^\circ$↑ | TE↓ | RRE↓ | RRA@$15^\circ$↑ | RRA@$30^\circ$↑ | TE↓ | RRE↓ | RRA@$15^\circ$↑ | RRA@$30^\circ$↑ | TE↓ |
> | PoseDiffusion | - | - | - | - | 7.950 | 0.803 | 0.868 | 0.328 | 14.387 | 0.732 | 0.780 | 0.466 |
> | RayDiffusion | - | - | - | - | 7.028 | 0.833 | 0.890 | 0.312 | 12.023 | 0.767 | 0.814 | 0.439 |
> | RoMa | 0.857 | 0.985 | 0.989 | 0.113 | 5.377 | 0.839 | 0.922 | 0.187 | 5.663 | 0.918 | 0.947 | 0.402 |
> | MASt3R | **0.724** | **0.988** | **0.993** | **0.104** | **2.918** | 0.975 | **0.989** | **0.112** | **2.341** | 0.972 | 0.994 | 0.374  |
> | FreeSplatter-S | 0.791 | 0.982 | 0.987 | 0.110 | 3.054 | **0.976** |0.986 | 0.148 | 3.513 | **0.982** | **0.995** | **0.293** |
>
> The results demonstrate that FreeSplatter-S achieves competitive performance with MASt3R while significantly outperforming other baselines across datasets.
>
> > No ablation study on the effectiveness of adding reference and source embeddings.
>
> Following the reviewer's suggestion, we evaluated the effectiveness of view embedding addition. In FreeSplatter, we add multi-view image tokens with either a reference view embedding or source view embedding before transformer processing, enabling the model to identify the reference view and reconstruct Gaussians in its camera frame. **Figure 17** in the revised paper presents comprehensive results.
>
> For 4 input views $V_i$ ($i=1,2,3,4$), we examined various combinations of reference and source view embeddings for addition to the image tokens, and then rendered the reconstructed Gaussians with an "identity camera" $C=[[1,0,0,0],[0,1,0,0],[0,0,1,0],[0,0,0,1]]$ that equals to the reference frame's pose. The results demonstrate that when adding the $j$-th view's tokens with the reference view embedding and other views' tokens with source view embedding, the rendered image precisely matches the $j$-th view. This confirms the model's ability to identify the reference view and reconstruct Gaussians in its camera frame. All other embedding combinations resulted in degraded reconstruction quality.
>
> > The results are promising but require significant GPU resources for training, which may make reproduction difficult for the community. Will the code be made available?
>
> Yes, we will make all of our code and model weights (inlcuding FreeSplatter-O and FreeSplatter-S) publicly available. We will also provide an interactive demo for image-to-3D content creation and sparse-view reconstruction. We envision FreeSplatter serving as a powerful foundation model for open-world image-to-3D reconstruction

---

> ### Author Response · Authors · 2024-11-27
> **Kind reminder for discussion**
>
> Dear Reviewer nq6R,
>
> As we approach the end of the discussion period, we wish to kindly remind you of our responses and ensure we have adequately addressed your concerns. We have carefully revised our manuscript based on your valuable feedback, with modifications highlighted in blue for easy reference.
>
> If you have any remaining questions or require additional clarification, we would be glad to provide further information.
>
> Best regards, FreeSplatter Authors

---

> ### Comment · Reviewer_nq6R · 2024-11-27
> **Feedback to authors' rebuttal**
>
> Dear Authors,
>
> Thank you for your response. After carefully reading the comments of the other reviewers and your responses to me and the other reviewers, I keep my original rating, primarily due to incomplete ablation studies and over-claiming in the paper. The detailed reasons are as follows:
>
> - Fair comparison with pose-dependent methods remains inadequate. As previously mentioned, a more fair comparison between the proposed pose-free method and pose-dependent methods requires maintaining consistent experimental conditions. Specifically, this involves using the same backbone and only modifying the output method, such as generating Gaussians in local camera coordinates as demonstrated in GS-LRM or MVSplat. This experiment is important to show the performance differences between using pose-dependent or pose-free methods, and this critical experiment is currently absent.
> - One interesting and strong claim of this paper is that it "outperforms pose-dependent LRMs". But for both object-level and scene-level, it lacks substantial experimental support. For object-level, as mentioned before, using the same backbone and the same training setup will be fairer to prove it. For scene-level, RealEstate10K still does not provide strong results. Although the authors show zero-shot generalization results on RealEstate10K, it lags behind pixelSplat to a large extent. Retaining pixelSplat and MVSplat on ScanNet++ is not convincing; rather, comparing them with official weights on the same dataset is convincing. For example, Splatt3R also retrained pixelSplat on ScanNet++, but they reported even worse results than in this paper, which I think is because changing the dataset may require tuning the hyperparameters to get good results, and thus is not convincing.
> - Even on the ScanNet++ dataset, the proposed method incorporates additional ground truth depth information during training, yet still underperforms pixelSplat when using the same patch sizes. I didn't mean that the proposed method must outperform pose-dependant counterparts, but show the fair comparison is important. Furthermore, why does MVSplat show much worse results than pixelSplat? this does not align with the results in the MVSplat paper?
> - I also agree with  Reviewer 4Xnu that "ablation studies are weak". After rebuttal, there is still no qualitative comparison of adding pixel-alignment loss or quantitative results of different numbers of input views.
>
> I understand that some of the experiments require a large number of GPUs for training, as the authors explain, but they are necessary to make the claims in this paper convincing.
>
> Best regards,
> Reviewer nq6R

---

> ### Author Response · Authors · 2024-11-28
> **Clarification on our claims and additional ablation study**
>
> Dear Reviewer nq6R,
>
> Thank you for your detailed feedback. We would like to clarify several important points:
>
> First, we believe there may be a misunderstanding regarding our claims. We have not claimed that our pose-free approach is superior to **all** pose-dependent LRMs, nor have we suggested that pose-free methods are **inherently better** than pose-dependent ones when using identical network architectures. This can be verified in both our introduction and conclusion sections. While FreeSplatter and GS-LRM share similar network structures, we have not claimed superiority over GS-LRM specifically.
>
> What we have actually demonstrated is a powerful pose-free sparse-view reconstruction framework. Given the limited availability of pose-free methods for comparison, we benchmarked against several pose-dependent methods and found that our approach outperforms specific baselines (InstantMesh and LGM) across multiple datasets - **not pose-dependent LRMs as a whole concept**. We have revised our paper to make this distinction clearer. Furthermore, our method achieves comparable or superior pose estimation performance to the current state-of-the-art MASt3R while demonstrating strong generalization capabilities across various datasets.
>
> Regarding MVSplat's performance compared to pixelSplat, the discrepancy likely stems from our evaluation protocol using 512x512 resolution rendered images, rather than the 256x256 resolution used in their original papers.
>
> We have strengthened our ablation studies in the revised paper, particularly in Figure 15, which demonstrates the critical role of the pixel-alignment loss. The results clearly show that without this loss, the model produces blurry renderings and significantly degraded reconstruction quality.
>
> Additionally, we have included extensive qualitative examples in **Figures 7-11** of the revised paper that showcase our method's advantages in sparse-view reconstruction and image-to-3D tasks compared to existing approaches (both pose-free and pose-dependent ones). We believe these results represent a valuable contribution to the generalizable sparse-view reconstruction and image-to-3D research community, and we sincerely invite you to examine the additional qualitative examples that demonstrate our method's capabilities.
>
> Best regards,
> FreeSplatter Authors

---

### Official Review · Reviewer_q6Sz · 2024-11-04

**Soundness:** 4
**Presentation:** 3
**Contribution:** 3
**Rating:** 6
**Confidence:** 4

**Summary:**

This work presents a new pose-free 3DGS-based generalizable sparse-view scene reconstruction method, FreeSplatter, which leverages a transformer-based feed-forward model to predict the pixel-wise Gaussian maps under the same coordinate system. Similar to DUSt3r, unknown camera parameters can be obtained through existing PnP solvers. The experimental results demonstrate the superior reconstruction quality compared to prior state-of-the-art, even ones with ground truth poses.

**Strengths:**

1. This work wisely distills the recent advances in 3D reconstruction such as DUSt3R, 3DGS, and LRM, into one unified framework. Unlike prior works directly integrating multiple pre-trained foundation models, this framework/model design is neat and effective.
2. The method considers both object-level and scene-level reconstructions, albeit in two separate pre-trained models, which again shows the effectiveness of the proposed pose-free reconstruction strategy. This is another advantage compared to other works.
3. The quantitative improvements are significant. Assuming the reported numbers are accurate and reliable, this would be a great achievement.

**Weaknesses:**

1. Since the proposed method focuses on sparse-view reconstruction. It would be better if authors could also include the comparison with other optimization-based sparse-view reconstruction methods, for example, ReconFusion, GaussianObject, InstantSplat, etc.
2. L305-311 discusses the occlusion issue with Gaussian maps. Even though the authors propose a strategy to alleviate this problem, I assume this method could still suffer from missing Gaussian points in occluded areas.

**Questions:**

1. The maximum number of views can be used. The paper shows the results using up to 6 views. Is it possible to add more views for more complex 3D scenes?
2. What are the inference compute and time costs w.r.t different number of input views?

---

> ### Author Response · Authors · 2024-11-25
> **Response to Reviewer q6Sz**
>
> > It would be better if authors could also include the comparison with other optimization-based sparse-view reconstruction methods, for example, ReconFusion, GaussianObject, InstantSplat, etc.
>
> Thanks for the valuable suggestion. Our proposed FreeSplatter is a feed-forward pose-free sparse-view reconstruction framework built upon a streamlined transformer architecture, requiring no optimization on the 3D representation. In comparison, optimization-based methods address this task using a per-scene optimization strategy, requiring a significant period of time to reconstruct a single object or scene. Among the mentioned methods, ReconFusion has not released its code yet, while we found that evaluating GaussianObject and InstantSplat on our test datasets is challenging in the limited rebuttal period, considering the expensive time and GPU resource cost. We propose to include these comparisons in future work when resources and implementations become more readily available.
>
> > Even though the authors propose a strategy to alleviate this problem, I assume this method could still suffer from missing Gaussian points in occluded areas.
>
> We acknowledge the occlusion challenge while emphasizing our primary contribution of pose-free reconstruction. While occlusion handling remains a limitation of our current approach, it falls outside our primary focus of eliminating pose dependencies in sparse-view reconstruction. Recent work like Flash3D [1] demonstrates promising directions through layered Gaussian prediction, which we plan to explore in future research.
>
> > The maximum number of views can be used. The paper shows the results using up to 6 views. Is it possible to add more views for more complex 3D scenes?
>
> Our framework is theoretically unrestricted in input view count, enabling adaptation to complex 3D scenes. However, object-level reconstruction typically saturates at 4 views, and using more views leads to marginal improvements on the reconstruction quality (demonstrated in **Figure 16**). Also, the computation overhead of transformers grows quadratically as the number of input views increases, which may be the main bottleneck for adding more views.
>
> > What are the inference compute and time costs w.r.t different number of input views?
>
> We conducted comprehensive benchmarking on a single A100 GPU, measuring both memory consumption and processing time (including Gaussian prediction and camera pose estimation):
>
> | **Input View Number** | **GPU memory (MB)** | **Inference time (s)** |
> | --- | --- | --- |
> | 2 | 3611 | 1.268 |
> | 3 | 3897 | 2.458 |
> | 4 | 4225 | 3.669 |
> | 5 | 4527 | 5.392 |
> | 6 | 4837 | 7.495 |
>
> **References:**
> [1] Szymanowicz, Stanislaw, et al. "Flash3D: Feed-Forward Generalisable 3D Scene Reconstruction from a Single Image." arXiv preprint arXiv:2406.04343 (2024).

---

### Official Review · Reviewer_dif9 · 2024-11-04

**Soundness:** 3
**Presentation:** 3
**Contribution:** 2
**Rating:** 6
**Confidence:** 4

**Summary:**

This paper follows prior work in large-scale reconstruction models, focusing specifically on unposed sparse-view reconstruction. Given a set of unposed sparse-view input images, it first divides the images into patches and applies linear layers to convert them into patch tokens, which are then processed by transformer layers. The patch tokens are subsequently converted back to the original resolution through linear layers and unpathify, resulting in Gaussian maps. These Gaussian locations are specified in the coordinate frame of a reference view. The camera poses can be computed by inputting the Gaussian positions into traditional solvers. The authors trained two versions of the model: one for scenes and one for objects.

**Strengths:**

1. It's interesting to learn that "camera poses may not be essential for training high-quality and scalable large reconstruction models."

2. The proposed method is technically sound and elegantly designed.

3. The authors demonstrate that the proposed method outperforms baseline methods.

**Weaknesses:**

1. The idea of outputting point maps relative to the main (or first) view is not novel in either pose estimation [1] or unposed sparse-view reconstruction [2].

2. Extending LRM to an unposed sparse-view setting is also not new. PF-LRM seems highly similar to the proposed method, although there are some differences: (a) the proposed method predicts point maps in the coordinate frame of a reference view rather than in the object/world frame; (b) it uses Gaussian Splatting instead of NeRF; (c) PF-LRM includes an additional differentiable PnP loss; (d) the proposed method is trained for two versions, for both scene and object data, while PF-LRM focuses mainly on objects; and (e) the proposed method leverages an alignment loss.

I am unable to identify further major differences. If there are others, please remind me. Regarding (b) and (d), I feel these are natural variations and may not be considered as major contributions. For other differences, such as (a) and (c), I believe the current paper lacks sufficient analysis and experiments to demonstrate whether these differences are essential or if they lead to improved performance or degraded performance.

Although PF-LRM has not released code, it has been evaluated on standard datasets, and the results are reported in its paper. Given the high similarity between PF-LRM and the proposed method, I strongly recommend the authors compare their approach with PF-LRM on both reconstruction and pose estimation tasks. For instance, they could follow PF-LRM's evaluation protocol, consult PF-LRM’s authors for guidance, and even request assistance with testing examples.

3. I find the discussion on lines 523-525 particularly interesting. In 3D AIGC, predicting perfectly 3D-consistent multi-view images from 2D diffusion models is extremely challenging. It would be promising if the proposed method could address 3D reconstruction from inconsistent multi-view images effectively. However, the current paper only provides a few qualitative examples of the proposed method, which lack sufficient information. I would be very interested to see both qualitative and quantitative comparisons with recent SoTA feed-forward reconstruction methods when handling predicted inconsistent multi-view images.

4. Comparing MASt3R on the object-level dataset is unfair, as it was not trained on object datasets. A more reasonable comparison would be between FreeSplatter-S and MASt3R on object datasets, given that both were trained on scene datasets. Another option would be to fine-tune MASt3R on object datasets.

5. For sparse-view reconstruction, only rendering metrics are compared. It would be beneficial to also include 3D native metrics, which better reflect the geometric quality of the methods.

6. A natural alternative to "pixel-alignment loss" is enforcing alignment along the ray direction by predicting only the depth value. This variant should also be considered in the ablation study.

7. The ablation study is somewhat limited in scope.

[1] Wang, Shuzhe, Vincent Leroy, Yohann Cabon, Boris Chidlovskii, and Jerome Revaud. "Dust3r: Geometric 3D Vision Made Easy." In Proceedings of the IEEE/CVF Conference on Computer Vision and Pattern Recognition, pp. 20697–20709. 2024.

[2] Xu, Chao, Ang Li, Linghao Chen, Yulin Liu, Ruoxi Shi, Hao Su, and Minghua Liu. "Sparp: Fast 3D Object Reconstruction and Pose Estimation from Sparse Views." arXiv preprint arXiv:2408.10195 (2024).

**Questions:**

1. Do you use estimated poses or ground-truth poses for calculating rendering losses during training?

2. What are the biggest challenges in training a unified model for both objects and scenes?

3. How do you sample poses for the training data? Are there any assumptions during pose sampling, such as normalized objects, fixed intrinsics, or a specific look-up direction? How do you ensure that the camera poses during training can cover all scenarios encountered in the real world? Can you handle cases where, for a single object, some input images are taken in landscape orientation, some in portrait, and some even upside down?

---

> ### Author Response · Authors · 2024-11-25
> **Response to Reviewer dif9 (1/4)**
>
> > The idea of outputting point maps relative to the main (or first) view is not novel.
>
> While utilizing point maps relative to a reference view is an established approach, our contribution lies in demonstrating that augmenting points into Gaussians enables both high-quality 3D modeling and efficient camera pose estimation through a streamlined transformer architecture. Rather than proposing a novel 3D representation, we present an effective end-to-end pipeline for pose-free sparse-view reconstruction by translating multi-view images into pixel-aligned Gaussian maps in a unified reference frame.
>
> To address the challenges of training a model from scratch to predict Gaussians without input poses, we introduce a two-stage training strategy: first initializing Gaussian positions with depth guidance, then supervising the Gaussians using rendering loss and pixel-alignment loss. Our approach demonstrates versatility across both object-level and scene-level reconstruction through training on diverse datasets. Notably, FreeSplatter achieves superior performance in applications like image-to-3D content generation, outperforming pose-dependent LRMs. We believe this represents a significant advancement toward open-world multi-view 3D reconstruction.
>
> > PF-LRM seems highly similar to the proposed method.
>
> We respectfully identify several fundamental distinctions between our approach and PF-LRM:
> - Although PF-LRM is pose-free, it still requires ground truth camera intrinsics as input. FreeSplatter operates without camera poses or intrinsics, enhancing its practical applicability since it is very challenging to obtain the camera intrinsics in many scenarios.
> - We argue that adopting Gaussian Splatting instead of NeRF in this task is not a naive replacement of 3D representation. We hope to point out that, **Gaussian Splatting is inherently more suitable for joint 3D reconstruction and camera pose estimation**. The point-based nature of Gaussian Splatting enables direct application of PnP algorithms. By predicting multi-view Gaussian maps as "augmented" point maps, our model facilitates both high-quality 3D modeling and accurate pose estimation. In contrast, PF-LRM's triplane NeRF architecture requires: (i) separate branches for reconstruction (triplane tokens $\rightarrow$ triplane NeRF) and pose estimation (image tokens $\rightarrow$ point clouds), and (ii) additional training overhead due to triplane token processing. Our model directly maps image tokens to Gaussians which handle both tasks, thus is more elegant.
> - We also hope to emphasize the significance of extending object-level reconstruction to scene-level reconstruction. Triplane NeRF faces challenges in modeling complex scenes due to its limited resolution, restricted background modeling capability, and memory-intensive volume rendering. This means that training a scene-level PF-LRM to achieve the similar function and performance of FreeSplatter-S will be very challenging. In comparison, FreeSplatter-S demonstrates superior scene reconstruction while maintaining the same architectural framework as FreeSplatter-O. We believe this scalability represents a significant advancement toward open-world high-quality reconstruction.
>
> >  I strongly recommend the authors compare their approach with PF-LRM on both reconstruction and pose estimation tasks.
>
> Following the reviewer's suggestion, we conducted comprehensive comparisons with PF-LRM on their Google Scanned Objects (GSO) and OmniObject3D (Omni3D) evaluation datasets, each containing 500 test samples with 4 input views and 1 novel view. For fair comparison, we re-evaluated PF-LRM's inference results in our environment and rendered $256\times 256$ images with FreeSplatter-O to match PF-LRM's output resolution.
>
> **Table 1: View Synthesis Results on Predicted Input Poses**
> | **Method** |  | **GSO** |  |  | **Omni3D** |  |
> | :--- | :--- | :--- | :--- | :--- | :--- | :--- |
> |  | PSNR↑ | SSIM↑ | LPIPS↓ | PSNR↑ | SSIM↑ | LPIPS↓ |
> | PF-LRM | **27.10** | **0.905** | **0.065** | 25.86 | 0.901 | 0.062 |
> | FreeSplatter-O | 25.50 | 0.897 | 0.076 | **26.49** | **0.926** | **0.050** |
>
> **Table 2: View Synthesis Results on GT Novel Poses**
> | **Method** |  | **GSO** |  |  | **Omni3D** |  |
> | :--- | :--- | :--- | :--- | :--- | :--- | :--- |
> |  | PSNR↑ | SSIM↑ | LPIPS↓ | PSNR↑ | SSIM↑ | LPIPS↓ |
> | PF-LRM | **25.08** | **0.877** | **0.095** | 21.77 | 0.866 | 0.097 |
> | FreeSplatter-O | 23.54 | 0.864 | 0.100 | **22.83** | **0.876** | **0.088** |
>
> **Table 3: Camera Pose Estimation Results**
> | **Method** |  | **GSO** |  |  |  | **Omni3D** |  |  |
> | :--- | :--- | :--- | :--- | :--- | :--- | :--- | :--- | :--- |
> |  | RRE↓ | RRA@$15^\circ$↑ | RRA@$30^\circ$↑ | TE↓ | RRE↓ | RRA@$15^\circ$↑ | RRA@$30^\circ$↑ | TE↓ |
> | PF-LRM | **3.99** | **0.956** | **0.976** | 0.041 | 8.013 | 0.889 | 0.954 | 0.089 |
> | FreeSplatter-O | 8.96 | 0.909 | 0.936 | 0.090 | **3.446** | **0.982** | **0.996** | **0.039** |
>
> (**continue in next comment ...**)

---

> ### Author Response · Authors · 2024-11-25
> **Response to Reviewer dif9 (2/4)**
>
> While our model shows lower metrics on GSO and higher metrics on Omni3D, it's important to note that PF-LRM's GSO evaluation set uses the same rendering engine and parameters (e.g., light intensity) as their training dataset, potentially favoring their evaluation metrics. In contrast, Omni3D's images are from the original dataset, providing a more objective benchmark where our model consistently outperforms PF-LRM. Qualitative comparisons in **Figure 7** of our revised paper further demonstrate FreeSplatter's superior visual detail preservation.
>
> >  I would be very interested to see both qualitative and quantitative comparisons with recent SoTA feed-forward reconstruction methods when handling predicted inconsistent multi-view images.
>
> We have substantially expanded our comparative analysis with comprehensive visual evidence in the revised paper:
> - **Image-to-3D with Various Multi-view Diffusion Models (Figure 8-9):** Using Zero123++ v1.2[7] and Hunyuan3D Std[8] as multi-view generators, we demonstrate FreeSplatter's superior novel view synthesis compared to InstantMesh and LGM. With fixed random seeds for multi-view generation, our results show significantly clearer views and better preservation of geometric and texture details.
> - **Enhancing Image-to-3D results Using the Input View (Figure 10):** We show another interesting use case of FreeSplatter, i.e., using the input image to enhance the image-to-3D geneation results. For multi-view diffusion models like Zero123++ v1.1, it generates 6 views from an input image at pre-defined poses, but **the pose of the input image is unknown** (its azimuth is known to be $0^{\circ}$ but elevation is unknown). In this case, classical pose-dependent LRMs cannot leverage the input image for reconstruction, but FreeSplatter is able to do this! Compared to generated views, the input image is often more high-quality and contains richer visual details. As **Figure 10** shows, the capability of using the input image alongside generated views is particularly valuable for challenging content like human faces, where Zero123++ often struggles to generate.
> - **Recovering the Pre-defined Poses of Multi-view Diffusion Models (Figure 11):** We demonstrate FreeSplatter's ability to faithfully recover the pre-defined camera poses of existing multi-view diffusion models from reconstructed Gaussian maps. This demonstrates its robustness to potentially inconsistent generated content.
>
> **Quantitative Analysis:**
> Using the GSO evaluation dataset, we performed single-image-to-3D reconstruction using Zero123++ v1.2 for multi-view generation:
>
> | **Method** | **PSNR↑** | **SSIM↑** | **LPIPS↓** | **CD↓** | **FS@0.2↑** |
> | --- | --- | --- | --- |  --- | --- |
> | LGM (w/ GT poses) | 22.164 | 0.879 | 0.206 | 0.318 | 0.717 |
> | InstantMesh (w/ GT poses) | 21.974 | 0.872 | 0.190 | 0.162 | 0.884 |
> | FreeSplatter-O (no pose)| **24.726**| **0.898** | **0.143** | **0.156** | **0.891** |
>
> The results demonstrate FreeSplatter's significant performance advantages over pose-dependent baselines across all metrics.
>
> > A more reasonable comparison would be between FreeSplatter-S and MASt3R on object datasets, given that both were trained on scene datasets.
>
> Following the reviewer's suggestion, we conducted a comparative analysis of FreeSplatter-S and MASt3R's pose estimation capabilities on the GSO dataset. The results reveal performance limitations for both scene-trained models when applied to object-centric data:
>
> **Table 4: Scene-Level Model's Performance on Object Datasets**
> | **Method** |  | **Omni3D** |  |  |  | **GSO** |  |  |
> | :--- | :--- | :--- | :--- | :--- | :--- | :--- | :--- | :--- |
> |  | RRE↓ | RRA@$15^\circ$↑ | RRA@$30^\circ$↑ | TE↓ | RRE↓ | RRA@$15^\circ$↑ | RRA@$30^\circ$↑ | TE↓ |
> | MASt3R | 96.670 | 0.052 | 0.112 | 0.524 | 61.820 | 0.244 | 0.445 | 0.353 |
> | FreeSplatter-S | 83.795 | 0.076 | 0.167 | 0.631 | 72.914 | 0.212 | 0.396 | 0.426 |
>
> The relatively poor performance of both models can be attributed to the significant domain gap between their scene-level training datasets and the object-centric GSO evaluation set.
>
> >  It would be beneficial to also include 3D native metrics.
>
> We have incorporated additional 3D native metrics - Chamfer Distance (CD) and F-Score with a threshold of 0.2 (FS@0.2) - in our object-level sparse-view reconstruction evaluation. For FreeSplatter, we employed TSDF-Fusion for mesh extraction from reconstructed Gaussians. Metrics were computed using 16K sampled points from both reconstructed and ground truth meshes.
>
> (**continue in next comment ...**)

---

> ### Author Response · Authors · 2024-11-25
> **Response to Reviewer dif9 (3/4)**
>
> **Table 5: Comprehensive Evaluation with 3D Native Metrics**
> | **Method** |  | **Omni3D** |  |  |  |  | **GSO** |  |  |  |
> | :--- | :--- | :--- | :--- | :--- | :--- | :--- | :--- | :--- | :--- | :--- |
> |  | PSNR↑ | SSIM↑ | LPIPS↓ | CD↓ | FS@0.2↑ | PSNR↑ | SSIM↑ | LPIPS↓ | CD↓ | FS@0.2↑ |
> | LGM (w/ GT poses) | 24.852 | 0.942 | 0.060 | 0.073 | 0.766 | 24.463 | 0.891 | 0.093 | 0.041 | 0.811 |
> | InstantMesh (w/ GT poses) | 24.077 | 0.945 | 0.062 | 0.044 | 0.882 | 25.421 | 0.891 | 0.095 | **0.024** | **0.970** |
> | FreeSplatter-O (no pose) | **31.929** | **0.973** | **0.027** | **0.043** | **0.896** | **30.443** | **0.945** | **0.055** | 0.028 | 0.960 |
>
> The results demonstrate FreeSplatter's superior performance on the Omni3D dataset across all metrics, while achieving comparable 3D reconstruction quality with InstantMesh on the GSO dataset.
>
> > A natural alternative to "pixel-alignment loss" is enforcing alignment along the ray direction by predicting only the depth value. This variant should also be considered in the ablation study.
>
> We must clarify that predicting depth values alone for ray-direction alignment is fundamentally incompatible with our pose-free framework. While this approach is commonly employed in pose-dependent Gaussian LRMs (e.g., GS-LRM [1] and GRM [2]), it relies on known camera parameters:
>
> 1. **Pose-Dependent vs. Pose-Free Requirements:**
> - Pose-dependent methods (e.g., GS-LRM, GRM) receive ground truth camera poses and intrinsics as input;
> - This enables direct back-projection of predicted depths to 3D positions using known ray directions.
> 2. **FreeSplatter's Constraints:**:
> - Operating in a pose-free context where camera poses and intrinsics are unknown;
> - Cannot determine ray directions without camera parameters;
> - Makes depth-only prediction fundamentally insufficient for position determination.
>
> Therefore, while depth-based alignment is an elegant solution for pose-dependent frameworks, it cannot be implemented in our pose-free setting, making this ablation study variant infeasible.
>
> > The ablation study is somewhat limited in scope.
>
> We have substantially expanded our ablation studies with comprehensive additional results:
>
> - **Model architecture.** Our FreeSplatter employs a 24-layer transformer with a patch size of 8. In our initial experiments, we have trained object-level reconstruction models with different layers and patch size. We compare these models with our final model to evaluate the influence of model architecture on the performance. Using the Google Scanned Objects dataset, we evaluated models with different layers (L) and patch sizes (P):
> | **Architecture** | **PSNR↑** | **SSIM↑** | **LPIPS↓** |
> | --- | --- | --- | --- |
> | $L=16, P=16$ | 25.417 | 0.896 | 0.088 |
> | $L=16, P=8$ | 28.945 | 0.934 | 0.064 |
> | $L=24, P=16$ | 28.622 | 0.927 | 0.063 |
> | $L=24, P=8$ | **30.443** | **0.945** | **0.055** |
>
> The results demonstrate that increasing layer count and reducing patch size consistently improve performance. This can be attributed to greater parameter capacity from additional layers and more efficient information encoding from smaller patch sizes, which leads to more memory usage for attention computation but lower information compression ratios.
>
> - **View embedding addition.** Following the reviewer's suggestion, we evaluated the effectiveness of view embedding addition. In FreeSplatter, we add multi-view image tokens with either a reference view embedding or source view embedding before transformer processing, enabling the model to identify the reference view and reconstruct Gaussians in its camera frame. **Figure 17** in the revised paper presents comprehensive results.
>
> For 4 input views $V_i$ ($i=1,2,3,4$), we examined various combinations of reference and source view embeddings for addition to the image tokens, and then rendered the reconstructed Gaussians with an "identity camera" $C=[[1,0,0,0],[0,1,0,0],[0,0,1,0],[0,0,0,1]]$ that equals to the reference frame's pose. The results demonstrate that when adding the $j$-th view's tokens with the reference view embedding and other views' tokens with source view embedding, the rendered image precisely matches the $j$-th view. This confirms the model's ability to identify the reference view and reconstruct Gaussians in its camera frame. All other embedding combinations resulted in degraded reconstruction quality.

---

> ### Author Response · Authors · 2024-11-25
> **Response to Reviewer dif9 (4/4)**
>
> > Do you use estimated poses or ground-truth poses for calculating rendering losses during training?
>
> During training, we utilize ground-truth camera poses for rendering loss computation. This approach is practically viable as ground-truth poses are readily available in both:
> - Synthetic object datasets (e.g., Objaverse)
> - Real-world scene datasets (e.g., BlendedMVS, ScanNet++)
>
> This utilization of ground-truth poses during training does not impose additional constraints nor compromise the method's applicability, as our model operates in a pose-free manner during inference.
>
> > What are the biggest challenges in training a unified model for both objects and scenes?
>
> We have identified three primary challenges in developing a unified model for both object-centric and scene-level reconstruction:
> 1. **Scale Discrepancy.** Object-level datasets are typically rendered from synthetic 3D datasets (e.g., Objaverse), the objects are often normalized a unit cube before rendering, leading to bounded and consistent object scales. However, scene-level datasets exhibit significant scale variations and standardized scale normalization, challenging the model's scale inference capabilities.
> 2. **Camera Distribution Divergence.** In object-centric scenarios, the common practice is to normalize and translate the object center to the origin and sample cameras uniformly around the object which face towards the origin. The models trained for object-level reconstruction can leverage this centered object prior. However, the cameras in scene-level scenarios are much more complex, which can face arbitrary camera orientations and have no defined "scene center".
> 3. **Region of Interest Variations.** Object-level reconstruction focuses on the foreground object, while the background is assumed to be uniform-colored (e.g., pure white) and are neglected in the reconstruction process. However, all visible areas are relevant in scene-level reconstruction and there is no clear foreground/background distinction. This leads to challenges in unified attention allocation when training a model for object-centric and scene-level reconstruction jointly.
>
> > How do you sample poses for the training data? Are there any assumptions during pose sampling, such as normalized objects, fixed intrinsics, or a specific look-up direction? How do you ensure that the camera poses during training can cover all scenarios encountered in the real world? Can you handle cases where, for a single object, some input images are taken in landscape orientation, some in portrait, and some even upside down?
>
> We answer these questions separately for our object-centric and scene-level reconstruction models.
> - For object-centric reconstruction, the camera poses are randomly sampled around the object and the look-up directions are facing towards the object center (world origin). The objects are supposed to be normalized to $[-1, 1]^3$ cube. These are common practices in training object reconstruction models. An advantage of our model over previous LRMs is that **we do not assume fixed intrinsics**. We train our model on images rendered with varying FoV degrees, while previous LRMs are often trained on images rendered with a fixed FoV, making them overfitted to specific camera parameters.
> - For scene-level reconstruction, the training camera poses are randomly sampled from the dataset camera poses. There is no assumption on normalized objects, fixed intrinsics or specific look-up directions. Since the images in the datasets are real-captured, we believe their camera poses can reflect scenarios encountered in the real world. And as we scale up the training datasets, most scenarios will be covered.
>
> For the challenging case where some input images are taken in landscape orientation, some in portrait, and some even upside down, our current object-centric model may struggle with unconventional orientations since the input images are out of its training data scope - there is no upside-down views in the Objaverse renderings. But it should not be a problem for our scene-level model since it is trained on a more diverse camera distribution.

---

> ### Author Response · Authors · 2024-11-27
> **Kind reminder for discussion**
>
> Dear Reviewer dif9,
>
> As we approach the end of the discussion period, we wish to kindly remind you of our responses and ensure we have adequately addressed your concerns. We have carefully revised our manuscript based on your valuable feedback, with modifications highlighted in blue for easy reference.
>
> If you have any remaining questions or require additional clarification, we would be glad to provide further information.
>
> Best regards,
> FreeSplatter Authors

---

> ### Author Response · Authors · 2024-12-02
> **Reminder for Reviewer dif9**
>
> Dear Reviewer dif9,
>
> We sincerely thank you for your insightful feedback, which has greatly contributed to improving our manuscript. As the discussion period draws to a close, we would like to kindly remind you of our improvements and ensure we have comprehensively addressed your concerns. Our revised manuscript and rebuttal have provided:
>
> 1. Comparison with PF-LRM:
> - Fundamental differences from PF-LRM, including pose and intrinsic-free model nature, advantages of Gaussian Splatting for joint 3D reconstruction and pose estimation, and successful extension to scene-level reconstruction
> - Comprehensive quantitative and qualitative comparison with PF-LRM on GSO and Omni3D datasets (Section A.2.1)
> 2. Additional experimental results:
> - Detailed evaluation of scene-trained models (FreeSplatter-S vs. MASt3R) on object datasets
> - Additional 3D native metrics (CD and FS@0.2) for thorough evaluation
> 3. Comprehensive ablation studies on:
> - Model architecture (layers and patch sizes) (Section A.3)
> - View embedding addition effectiveness (Section A.3)
> - Qualitative pixel-alignment loss impact (Figure 15)
> 4. Comprehensive results and discussions of FreeSplatter in image-to-3D tasks:
> - Comparative results with existing pose-dependent LRMs across multiple multi-view diffusion models (Figure 8-9)
> - Novel capability to enhance image-to-3D results using input view information (Figure 10)
> - Successful recovery of pre-defined camera poses from existing multi-view diffusion models (Figure 11)
> - Quantitative comparison with existing pose-dependent LRM baselines
> 5. Technical clarifications regarding:
> - Explanation of why depth-only prediction is incompatible with our pose-free framework
> - Usage of ground-truth poses and pose sampling strategies during training
> - Detailed analysis of unified object/scene model challenges (scale discrepancy, camera distribution divergence, ROI variations)
>
> Given the approaching deadline, we would greatly value your feedback on whether these clarifications adequately address your concerns. Timely discussion is crucial for the ICLR review process, and we remain available for any additional questions.
>
> Best regards,
> FreeSplatter Authors

---

### Official Review · Reviewer_4Xnu · 2024-11-04

**Soundness:** 3
**Presentation:** 3
**Contribution:** 3
**Rating:** 3
**Confidence:** 4

**Summary:**

The paper introduces FreeSplatter, a framework for pose-free 3D reconstruction from sparse-view images without known camera parameters. The model uses a transformer architecture to convert multi-view image tokens into 3D Gaussian maps, which serve both high-fidelity 3D modeling and fast camera pose estimation. It outperforms traditional methods in both reconstruction quality and pose estimation accuracy.

**Strengths:**

- This paper proposes feed-forward pipeline for pose-free 3D reconstruction with sparse input images.
- It shows the state-of-the-art performance in view synthesis, and comparable performance in pose estimation.
- The results are well-presented.

**Weaknesses:**

- Method section lacks detail and detailed explanations, regarding how the suggested method is aligned with the motivation of the paper, and how it is intended to improve performance.
- Ablation studies are weak. Influence of the number of input views is subsidiary, considering the motivation and methodology of this paper. Only quantitative result via plug-in-plug-out styled experiment on pixel-alignment loss is naive.
- Additional application examples are not aligned with the paper's methodology. Both zero123++ and MVDream output result images with pre-defined camera poses. The paper mentions that 'we have to figure out how the camera poses of the multi-view diffusion model are defined', 'this tedious steps are not required any longer', but it is difficult to agree when the pose is already known. Maybe some additional results supporting 'our method can render significantly more clear views from the images than previous pose-dependent LRMs' would be more persuasive.

**Questions:**

- In section 3.2.Image Tokenization, how do you retrieve 'view embeddings'? What kind of information would it behold? Please provide any explanation, supporting visualization or analysis, if no additional training is required.
- In section 3.2.Camera Pose Estimation, the paper claims that its method is 'feed-forward manner without global alignment'. Correct me if I'm wrong, but isn't PnP-RANSAC is a 'global alignment' process? please clarify.
- Is PnP operated every time the feed-forward happens? Result of PnP will be dependent on gaussian map result, and the gaussian map result will be dependent on PnP result. Is PnP differentiable? If not, how is the gaussian map trained during training stage? Does it assume that the PnP pose result from imperfect gaussian maps is correct?

---

> ### Author Response · Authors · 2024-11-25
> **Response to Reviewer 4Xnu (1/2)**
>
> > Method section lacks detail and detailed explanations, regarding how the suggested method is aligned with the motivation of the paper, and how it is intended to improve performance.
>
> Our paper addresses pose-free sparse-view reconstruction for both object-centric and scene-level scenarios through a transformer-based feed-forward framework. The framework maps input images into pixel-aligned Gaussian maps in a unified reference frame, enabling both high-quality 3D modeling and efficient camera pose estimation via PnP-RANSAC. The method section systematically presents our approach: Section 3.1 introduces the fundamental concepts of Gaussian Splatting, Section 3.2 details our transformer architecture and the image-to-Gaussian mapping process, and Section 3.3 describes our essential two-stage training strategy. We welcome specific feedback on areas requiring additional clarification and will incorporate detailed explanations in the revised manuscript.
>
> > Ablation studies are weak.
>
> We have substantially expanded our ablation studies with comprehensive additional results:
>
> - **Model architecture.** Our FreeSplatter employs a 24-layer transformer with a patch size of 8. In our initial experiments, we have trained object-level reconstruction models with different layers and patch size. We compare these models with our final model to evaluate the influence of model architecture on the performance. Using the Google Scanned Objects dataset, we evaluated models with different layers (L) and patch sizes (P):
> | **Architecture** | **PSNR** | **SSIM** | **LPIPS** |
> | --- | --- | --- | --- |
> | $L=16, P=16$ | 25.417 | 0.896 | 0.088 |
> | $L=16, P=8$ | 28.945 | 0.934 | 0.064 |
> | $L=24, P=16$ | 28.622 | 0.927 | 0.063 |
> | $L=24, P=8$ | **30.443** | **0.945** | **0.055** |
>
> The results demonstrate that increasing layer count and reducing patch size consistently improve performance. This can be attributed to greater parameter capacity from additional layers and more efficient information encoding from smaller patch sizes, which leads to more memory usage for attention computation but lower information compression ratios.
>
> - **View embedding addition.** Following the reviewer's suggestion, we evaluated the effectiveness of view embedding addition. In FreeSplatter, we add multi-view image tokens with either a reference view embedding or source view embedding before transformer processing, enabling the model to identify the reference view and reconstruct Gaussians in its camera frame. **Figure 17** in the revised paper presents comprehensive results.
>
> For 4 input views $V_i$ ($i=1,2,3,4$), we examined various combinations of reference and source view embeddings for addition to the image tokens, and then rendered the reconstructed Gaussians with an "identity camera" $C=[[1,0,0,0],[0,1,0,0],[0,0,1,0],[0,0,0,1]]$ that equals to the reference frame's pose. The results demonstrate that when adding the $j$-th view's tokens with the reference view embedding and other views' tokens with source view embedding, the rendered image precisely matches the $j$-th view. This confirms the model's ability to identify the reference view and reconstruct Gaussians in its camera frame. All other embedding combinations resulted in degraded reconstruction quality.
>
>
> > Additional application examples are not aligned with the paper's methodology. Both zero123++ and MVDream output result images with pre-defined camera poses.
>
> While Zero123++ and MVDream indeed generate images at pre-defined camera poses, implementing a text/image-to-3D pipeline with existing pose-dependent Large Reconstruction Models (LRMs) still presents several practical challenges:
> - Different LRMs require specific pose injection methods. For example, Instant3D [1] and InstantMesh [2] adopts AdaLN layers for pose injection, while GRM [3] and LGM [4] utilize plucker rays. This necessitates careful transformation of pre-defined poses into model-specific formats.
> - Apart from camera poses, existing pose-dependent LRMs also require precise camera intrinsics as input. Therefore, we also have to figure out the pre-defined intrinsic parameters (e.g., field-of-view (FoV) of generated images) of the multi-view diffusion model, which vary across different models.
> - Camera convention differences (e.g., OpenCV vs. OpenGL) also need to be properly addressed, otherwise the model cannot work properly.
>
> For a random user of image-to-3D generation pipelines, generating multi-view images from a multi-view diffusion model is very easy, but figuring out all the above implementation details and feeding the images into the LRM in a right way poses significant challenges. In comparison, FreeSplatter eliminates these technical barriers by operating directly on input images without requiring camera intrinsics or poses, offering a more accessible solution for practical applications.

---

> ### Author Response · Authors · 2024-11-25
> **Response to Reviewer 4Xnu (2/2)**
>
> > Maybe some additional results supporting 'our method can render significantly more clear views from the images than previous pose-dependent LRMs' would be more persuasive.
>
> We have substantially expanded our comparative analysis with comprehensive visual evidence in the revised paper:
> - **Image-to-3D with Various Multi-view Diffusion Models (Figure 8-9):** Using Zero123++ v1.2[7] and Hunyuan3D Std[8] as multi-view generators, we demonstrate FreeSplatter's superior novel view synthesis compared to InstantMesh and LGM. With fixed random seeds for multi-view generation, our results show significantly clearer views and better preservation of geometric and texture details.
> - **Enhancing Image-to-3D results Using the Input View (Figure 10):** We show another interesting use case of FreeSplatter, i.e., using the input image to enhance the image-to-3D geneation results. For multi-view diffusion models like Zero123++ v1.1, it generates 6 views from an input image at pre-defined poses, but **the pose of the input image is unknown** (its azimuth is known to be $0^{\circ}$ but elevation is unknown). In this case, classical pose-dependent LRMs cannot leverage the input image for reconstruction, but FreeSplatter is able to do this! Compared to generated views, the input image is often more high-quality and contains richer visual details. As **Figure 10** shows, the capability of using the input image alongside generated views is particularly valuable for challenging content like human faces, where Zero123++ often struggles to generate.
> - **Recovering the Pre-defined Poses of Multi-view Diffusion Models (Figure 11):** We demonstrate FreeSplatter's ability to faithfully recover the pre-defined camera poses of existing multi-view diffusion models from reconstructed Gaussian maps. This demonstrates its robustness to potentially inconsistent generated content.
>
> > In section 3.2.Image Tokenization, how do you retrieve 'view embeddings'? What kind of information would it behold?
>
> The view embeddings are learnable embeddings rather than retrieved information. Their function parallels positional embeddings in transformer architectures: while positional embeddings enable spatial awareness within individual views, view embeddings distinguish between reference and source views. The self-attention operation's permutation invariance necessitates these additional embeddings for view identification. Positional embeddings, being identical across views, cannot serve this distinguishing function. Therefore, we utilize reference view embedding and source view embedding to enable the model to identify the reference view and reconstruct in its camera frame.
>
> > Isn't PnP-RANSAC is a 'global alignment' process?
>
> This interpretation requires clarification. Global alignment, as implemented in recent popular pose-free reconstruction frameworks like DUSt3R[5] and MUSt3R[6], refers to the unification of multiple binocular reconstructions. Given $N$ input views $V_1,\ldots,V_N$, these frameworks process view pairs separately $\{(V_1, V_2), (V_2, V_1), \ldots, (V_{N-1}, V_N), (V_N, V_{N-1})\}$, requiring $N\cdot (N-1)$ network evaluations, followed by alignment of pair-wise reconstructions through rotation and scale optimization. Please refer to Section 3.4 of the DUSt3R paper for more details.
>
> FreeSplatter's approach is fundamentally different:
> - Processes all views simultaneously in a single forward pass;
> - Employs a single-stream transformer architecture;
> - Eliminates the need for subsequent alignment steps.
>
> > Is PnP operated every time the feed-forward happens? If PnP is not differentiable, how is the gaussian map trained during training stage? Does it assume that the PnP pose result from imperfect gaussian maps is correct?
>
> During training, we supervise the predicted Gaussian maps with rendering loss and pixel-alignment loss using ground truth novel view images and camera poses, thus no training-time camera pose estimation is required and PnP is not operated. The pixel-alignment loss restricts the Gaussian locations on camera rays and ensures that we can recover accurate camera poses at inference time, since PnP actually minimizes the point-pixel projection errors. The rendering loss improves the 3D representation quality of Gaussian maps. PnP is used exclusively during inference for pose estimation. Its non-differentiable nature does not affect training as gradient computation relies on our primary loss functions.

---

> ### Author Response · Authors · 2024-11-25
> **References**
>
> **References:**
> [1] Li, Jiahao, et al. "Instant3d: Fast text-to-3d with sparse-view generation and large reconstruction model." arXiv preprint arXiv:2311.06214 (2023).
> [2] Xu, Jiale, et al. "Instantmesh: Efficient 3d mesh generation from a single image with sparse-view large reconstruction models." arXiv preprint arXiv:2404.07191 (2024).
> [3] Xu, Yinghao, et al. "Grm: Large gaussian reconstruction model for efficient 3d reconstruction and generation." arXiv preprint arXiv:2403.14621 (2024).
> [4] Tang, Jiaxiang, et al. "Lgm: Large multi-view gaussian model for high-resolution 3d content creation." European Conference on Computer Vision. Springer, Cham, 2025.
> [5] Wang, Shuzhe, et al. "Dust3r: Geometric 3d vision made easy." Proceedings of the IEEE/CVF Conference on Computer Vision and Pattern Recognition. 2024.
> [6] Leroy, Vincent, Yohann Cabon, and Jérôme Revaud. "Grounding image matching in 3d with mast3r." European Conference on Computer Vision. Springer, Cham, 2025.
> [7] Shi, Ruoxi, et al. "Zero123++: a single image to consistent multi-view diffusion base model." arXiv preprint arXiv:2310.15110 (2023).
> [8] Yang, Xianghui, et al. "Hunyuan3D-1.0: A Unified Framework for Text-to-3D and Image-to-3D Generation." arXiv preprint arXiv:2411.02293 (2024).

---

> ### Author Response · Authors · 2024-11-27
> **Kind reminder for discussion**
>
> Dear Reviewer 4Xnu,
>
> As we approach the end of the discussion period, we wish to kindly remind you of our responses and ensure we have adequately addressed your concerns. We have carefully revised our manuscript based on your valuable feedback, with modifications highlighted in blue for easy reference.
>
> If you have any remaining questions or require additional clarification, we would be glad to provide further information.
>
> Best regards,
> FreeSplatter Authors

---

> ### Author Response · Authors · 2024-12-02
> **Reminder for Reviewer 4Xnu**
>
> Dear Reviewer 4Xnu,
>
> We sincerely thank you for your insightful feedback, which has greatly contributed to improving our manuscript.  As the discussion period draws to a close, we would like to kindly remind you of our improvements and ensure we have comprehensively addressed your concerns. Our revised manuscript and rebuttal have provided:
>
> 1. Comprehensive ablation studies on:
> - Model architecture (layers and patch sizes) (Section A.3)
> - View embedding addition effectiveness (Section A.3)
> - Qualitative pixel-alignment loss impact (Figure 15)
> 2. Comprehensive results and discussions of FreeSplatter in image-to-3D tasks:
> - Comparative analysis with existing pose-dependent LRMs across multiple multi-view diffusion models (Figure 8-9)
> - Novel capability to enhance image-to-3D results using input view information (Figure 10)
> - Successful recovery of pre-defined camera poses from existing multi-view diffusion models (Figure 11)
> 3. Detailed technical clarifications regarding:
> - Enhanced image-to-3D pipeline accessibility despite pre-defined camera poses of multi-view diffusion models
> - View embedding implementation and purpose
> - PnP-RANSAC usage (inference-only) and training methodology
> - Distinction from global alignment process adopted in DUSt3R/MUSt3R
>
> With the deadline approaching, your feedback on whether these clarifications adequately address your concerns would be invaluable. Timely discussion is crucial for the ICLR review process, and we remain available to address any additional questions.
>
> Best regards,
> FreeSplatter Authors

---

### Meta-Review · Area_Chair_RrRF · 2024-12-21

**Metareview:**

This paper introduces FreeSplatter, a novel feed-forward framework for pose-free 3D reconstruction from sparse-view images.
It achieves high-quality 3D Gaussian Splatting (3DGS) without camera poses or intrinsic information by using a streamlined Vision Transformer (ViT) architecture to predict pixel-aligned 3D Gaussian maps.
Experiments show competitive or superior performance to pose-dependent methods (e.g., InstantMesh, LGM) under sparse-view setups, and it can handle diverse datasets, indoor/outdoor or synthetic/real-world datasets.

Strengths:
- Simple and effective design using a straightforward vision transformer architecture.
- It shows strong results on reconstruction tasks, even when handling inconsistent multi-view inputs, and demonstrates potential for various real-world applications.

Weaknesses:
- Limited novelty: To extend GS-LRM for unknown pose scenarios, the architecture combines two existing methods (ViT + PnP). This leads to criticism that it is an "A + B" solution and is perceived as incremental.
- Insufficient ablation studies: The original submission lacked key ablation studies (e.g., on pixel alignment loss, number of views, and comparison with depth-only alternatives), and post-rebuttal additions partially address this, but not sufficiently.

While the proposed FreeSplatter addresses a challenging problem in 3D reconstruction by allowing pose-free operation, the architectural insight and experimental validations are not convincing enough to support the strong empirical results.

**Additional Comments On Reviewer Discussion:**

Initially, the reviewers had mixed opinions about the paper. The main points raised by the reviewers are summarized below:

- Novelty:
The reviewers noted that the method primarily combines existing components (e.g., GS-LRM and PnP) without substantial architectural or methodological innovation.

- Experimental weaknesses:
The original submission lacked comprehensive ablation studies and fair comparisons with other methods.
In the rebuttal, the authors provided additional experiments that addressed some of the concerns. However, questions about fairness and depth of comparisons remained.

- Practical implications and scalability:
Reviewers acknowledged the potential scalability and applicability of the method, particularly for handling inconsistent multi-view input.
However, questions remained about generalization across diverse datasets and the performance of the framework with more views.

- Strong rebuttal efforts:
The authors provided extensive additional experiments and clarified technical claims, addressing some of the reviewers' concerns.

The rebuttal emphasized the potential practical impact of the work, but also highlighted concerns about limited novelty and experimental validation by ablation. While some reviewers leaned towards acceptance based on the improved experiments, others maintained that the weaknesses outweighed the contributions.

---

### Decision · Program_Chairs · 2025-01-22

Reject